# Stochastic parametric skeletal dosimetry model for humans: Anatomical-morphological basis and parameter evaluation

Evgenia I. Tolstykh[1], Pavel A. Sharagin[1], Elena A. Shishkina[1,2], Alexandra Yu Volchkova[1], Michael A. Smith[3]*, Bruce A. Napier[3]

1 Urals Research Center for Radiation Medicine, Chelyabinsk, Russia, 2 Chelyabinsk State University, Chelyabinsk, Russia, 3 Pacific Northwest National Laboratory, Richland, Washington, United States of America

* Michael.Smith@pnnl.gov

**Editor:** Hesham M. H. Zakaly, Ural Federal University named after the first President of Russia B N Yeltsin Institute of Physics and Technology: Ural'skij federal'nyj universitet imeni pervogo Prezidenta Rossii B N El'cina Fiziko-tehnologiceskij institut, RUSSIAN FEDERATION

## Abstract

Radiation exposure of the hematopoietic system that results in a radiation dose to bone marrow of more than 100 mGy leads to an increase in the risk of leukemia in humans. Excess relative risk of leukemia was observed in cohorts whose members lived in the territories of the Southern Urals that were radioactively contaminated in the 1950s. As part of the dosimetric support of epidemiological studies of these cohorts, an original methodology for stochastic bone dosimetric modeling was developed, termed the Stochastic Parametric Skeletal Dosimetry (SPSD) model. The purpose of this work was to present the anatomical and morphological bases of the SPSD model, which includes an assessment of the parameters of the microstructure of the trabecular bone in the hematopoietic areas of the human skeleton, as well as a description of the macrostructural division (segmentation) of hematopoietic areas into simple bone segments. As a result, an anatomical-morphological basis of the SPSD model was created based on published data. Data collection work included the analysis of original articles, atlases, manuals, monographs and the formation of primary data files. Data on the duration of hematopoiesis in various parts of the skeleton; and data on age-related changes in the microstructure and linear dimensions of human bones and their segments were analyzed. The paper describes the full set of parameters to be used for the dosimetric model for newborns, children aged 1, 5 and 10 years, as well as for adolescents aged 15 years and adults; for the latter, sex differences in bone size were considered. In total, the SPSD model includes 289 unique basic (bone) phantom segments, each of which is described by 7 or more parameters describing the microstructure, thickness of the cortical layer and linear dimensions. Population variability was estimated for each parameter. The approach to SPSD modeling, i.e., the use of simple geometric shapes, was successfully verified using independent datasets on bone masses and volumes.

**Data availability statement:** All relevant data are within the manuscript and its Supporting Information files.

**Funding:** This work was funded by Federal Medical-Biological Agency of Russia (MOD, EIT, EAS, PAS, VIZ) and the U.S. Department of Energy's Office of International Health Programs (MAS, BAN) in the framework of joint US-Russia JCCRER Project 1.1 (https://www.energy.gov/ehss/russian-health-studies-program). The funders had no role in study design, data collection and analysis, decision to publish, or preparation of the manuscript.

**Competing interests:** The authors have declared that no competing interests exist.

## Introduction

Radiation exposure of the hematopoietic system results in an increase of leukemia risk in humans associated with radiation dose to bone marrow of more than 100 mGy [1,2]. Excess relative risk of leukemia was observed in cohorts [2–4], whose members lived in the territories of the Southern Urals. The territory was radioactively contaminated in the 1950s due to the activities of the Mayak Production Association (PA): the first weapons-grade plutonium production facility in the former Soviet Union. Bone-seeking beta-emitters $^{89,90}$Sr were the main source of radiation for the cohort members. These radionuclides accumulate in trabecular and cortical bone tissue (mineralized matrix) and locally irradiate the bone marrow located between the bone trabeculae.

Dosimetric support for epidemiological studies of the Urals cohorts requires an adequate assessment of bone marrow irradiation doses. Assessment of internal doses involves the use of radionuclide intake functions, as well as biokinetic and dosimetric models. The dosimetric models incorporate computational organ phantoms that estimate absorbed dose in the target organ. Most ICRP phantoms are based on limited biological material (e.g., single cadavers). This limitation does not allow for consideration of the contribution of individual variability of skeletal structures to the uncertainty of dose estimates. As part of the dosimetric foundation for epidemiological studies of Urals cohorts, an original method of stochastic bone dosimetry modeling has been developed; and a general description of this new dosimetry model was presented [5,6]. The model was developed to calculate the radiation doses to active marrow (target tissue) resulting from $^{89,90}$Sr deposited in the trabecular and cortical bone (source tissues). The sites of the bone where hematopoiesis occurs are hereinafter referred to as hematopoietic sites. The elaborated model was named Stochastic Parametric Skeletal Dosimetry [SPSD] model. The modeling includes parametric generation of computational bone segment phantoms for hematopoietic sites of the human skeleton in voxel representation.

The development and implementation of the SPSD approach is a complex, multi-stage study that cannot be fully covered within a single paper. In this regard, we prepared a series of the papers that describe various aspects of SPSD modeling. General approaches were described by Degteva et al. [5,6]; mathematical models and software were described in [7,8]; the current paper presents the description of the anatomical-morphological basis for the SPSD model; and another publication [9] summarizes the data on phantoms generated for reference ages. The methods for the evaluation of dose uncertainty and sensitivity were discussed in detail [10–12]; a paper on dose coefficients (final modeling results) is currently in preparation. To clarify the overall picture, we have included a flowchart of SPSD modeling (Fig 1). The current paper describes the methodology and results of the initial steps (Fig 1A–1D), as well as the place of these results in the overall scheme of SPSD modeling.

The first step (Fig 1A) is the analysis of AM location in different bone sites for persons of different ages. At birth all bone marrow can be considered as AM. With increasing age AM converts into inactive or yellow (fatty) marrow. For modeling purposes, the age of AM conversion should be determined for each specific

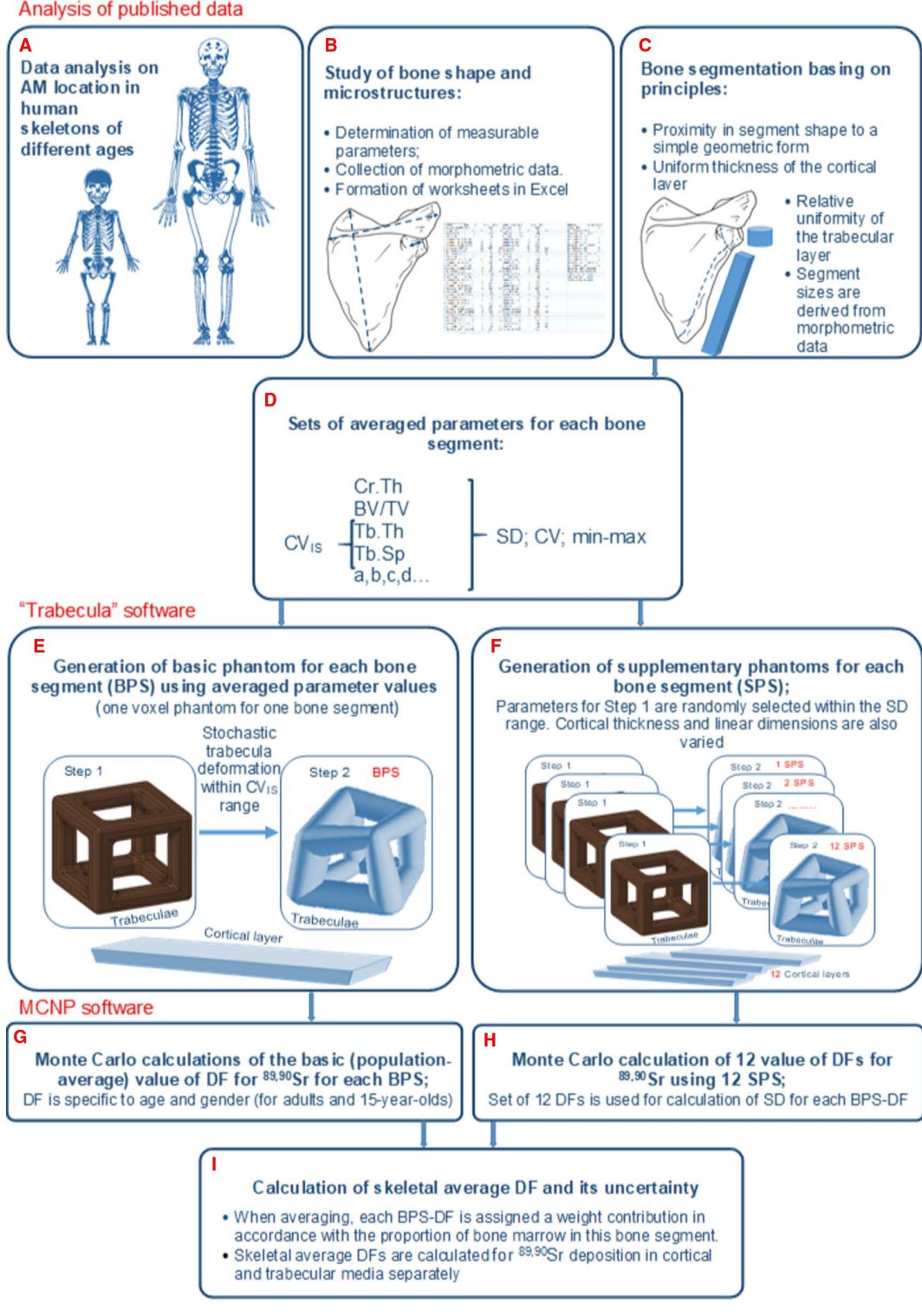

**Fig 1. Flowchart of SPSD-modeling.** Stages **A-C** are performed with use of published data: **A**- selection of skeleton bones with active marrow for reference ages; **B**- study of bone shapes, collection of morphometric parameters; (**C**) selection of bone segments with measured parameters and assignment of suitable geometric shapes to them (e.g., box, cylinder, cone). **D** – averaging and accepting parameters for bone segments and their geometric analogues (shapes): thickness of cortical layer (Ct.Th); trabecular thickness (Tb.Th), trabecular separation (Tb.Sp), fraction of bone volume in total

volume of spongiosa (BV/TV), bone segment dimensions (a,b,c,d…) and characteristics of parameter variability. Stages **E-F** are performed using "Trabecula" software for generation of parametric phantoms in voxel representation. **E – Step 1** – spongiosa model is generated as an isotropic 3-D-grid of rod trabeculae with bone marrow in the space between them; Ct.Th and other parameters correspond to population-averaged values for bone segment. **E- Step 2** – stochastic 3D-deformation of trabeculae (in the range of intra- specimen variability $CV_{is}$) is carried out to make a model equivalent (BPS) of a real trabecular structure. **F** – generation of a set of 12 supplementary phantoms (SPS), whose parameters for Step 1 are randomly selected in the range SD (d). Stages **G-H** are performed using MCNP software for Monte Carlo simulation of [89,90]Sr electron/photon transport in the generated voxel phantoms. **G**- calculation of the population-average values of DF for each BPS; **H**- calculation of DF for each supplementary phantom (12 DFs for each bone-segment), estimation of SD for BPS-DF. **I** – estimation of skeletal average DF and its uncertainty for each reference age for [89,90]Sr source in cortical and trabecular media separately.

hematopoietic site. The data on the AM distribution per large hematopoietic sites were described [13] and discussed [6]. The data are mainly based on the quantitative and qualitative histologic data on age-specific marrow cellularity for specific bones [14–18]. However, the described skeletal sites do not always match the bone segments created in the SPSD model. Therefore, one of the tasks of this study was to analyze the duration of hematopoiesis in large skeletal sites (mainly long tubular bones) using the latest magnetic resonance tomography (MRT) and positron emission tomography (PET) data available in the literature.

Detailed modeling of bony forms and trabecular structures in the SPSD model (Fig 1B, 1C) is necessary because of the short path length of the electrons in the [89,90]Sr emission spectrum. Assuming the continuous-slowing-down approximation is valid, the mean free path of electrons through the spongiosa (combining bone trabeculae and the inter-trabecular spaces occupied by marrow) is about 2 mm, and more than 99% of the energy would be absorbed within 10 mm. This means that the maximum path length in spongiosa is comparable to the linear dimensions of small bones and that some electrons escape from trabecular bone to cortical bone. To account for energy losses, the linear dimensions of bones must be considered in the dosimetry model.

Even though the trabecular structure of bone is now being extensively studied, we did not find published research summarizing these data, which would have allowed us to estimate the average values of spongiosa parameters in hematopoietic bone sites in people of different ages. The work is also complicated by the fact that the parameters of trabecular structures differ significantly in different bones. The analysis of the published data has shown that a weak correlation is observed between the parameters of spongiosa of various parts of the skeleton [19]. That is, each bone containing AM (each hematopoietic site) requires careful and individualized consideration.

The segmentation (Fig 1C) makes it possible to isolate areas of the bone that are sufficiently homogeneous in microstructure and cortical layer thickness; in addition, small segments allow for phantoms with high voxel resolution.

The objective of the current paper is to determine numerical values of the trabecular bone characteristics (Tb.Th; Tb.Sp; BV/TV), cortical thickness (Cr.Th) as well as bone dimensions for hematopoietic sites of a person aged 0, 1, 5, 10, 15, adults (> 25) years. To achieve the set goal, the following specific tasks were completed:

- Analysis of the age-dependent distribution of AM in the human skeleton, to determine a set of hematopoietic sites and bone segments suitable for the SPSD model;

- Assessment of the age-dependent microstructure of the trabecular bone in the hematopoietic sites of the human skeleton to quantify trabecular bone characteristics;

- Analysis of standard anatomical and morphological characteristics of bones containing AM to assess dimensional parameters of bones to facilitate their segmentation.

Therefore, the focus of the study is on methodological innovation and SPSD model parameter evaluation.

## Approaches to published data selection and analysis

We used the data from basic guidelines for osteology (e.g., [20,21]), electronic resources containing collections of X-ray images [22], and atlases on human anatomy (e.g., [23,24]). Published data from peer-reviewed journals were considered

for estimation of some specific bone parameters, along with data from doctoral theses. The search of published data was done using Internet search engines (e.g., Google, PubMed, Academia, e-library). Basic definitions that guided the analyses are presented in Table 1. The characteristics of the bone microstructure (Table 2) correspond to the standard histological nomenclature [25,26].

## Characteristics of subjects studied

Archetypical control population bone measurements were determined from study subjects. Study subjects must be described by investigators as healthy or without bone disease (asymptomatic bones). This is a prerequisite for most anthropological research. It is well known that there is a secular trend in the size of the human body (skeleton). These changes occurred in waves; however, since the end of the 19th century, human size has increased [27]. In our study, collected data on measurements of bone sizes are mainly related to persons who lived in the second half of the 20th and in the 21st century. To assess the parameters of the microstructure, the data obtained for people living in different periods of time were combined, because there is no reason to believe that the bone microstructure is subject to a secular trend. Several control population characteristics were of importance. Two different study subject parameter confidence weighting factors are also introduced, $W_N$ and $W_A$.

**Ethnicity of subjects.** Caucasians or peoples of greater Asian descent. The choice of ethnic groups is associated with the ethnic heterogeneity of the Urals cohorts, for which the SPSD model was developed. However, we did not carry out a quantitative weight estimation of parameters based on ethnicity.

**Table 1. Basic definitions.**

| Term | Definition |
|---|---|
| Bone marrow | Hematopoietic tissue located in bone cavities; includes hematopoietic cells at different stages of differentiation, connective tissue forming the stroma, fat cells and mature blood cells (leukocytes and erythrocytes). Depending on the proportion of fat cells, a distinction is made between red (active) bone marrow (AM) with a minimum number of fat cells and yellow (inactive) bone marrow (IM), where active hematopoiesis does not occur. Unlike children in the first year of life, the AM of an adult always contains a significant amount of adipose tissue. In the dosimetric model, AM is considered as the target tissue of radiation. |
| Cortical bone | Type of bone tissue with a high proportion of bone (mineralized) substance, which is more than 0.7, and on average 0.90–0.97. Non-mineralized areas are represented by Haversian canals with blood vessels, osteocyte lacunae with processes that form a thin lacunar network. In the dosimetric model, the volume of the cortical bone is considered as a source tissue of radiation. |
| Trabecular bone | Type of bone tissue with a low proportion of bone (mineralized) substance (trabeculae), which is less than 0.5, and on average from 0.1 to 0.3. The intertrabecular space is filled with bone marrow and blood vessels. In the dosimetric model, the volume of the trabecular bone is considered as a source tissue of radiation. |
| Segmentation | Dividing a bone into small homogeneous segments that can be described by a relatively small number of voxels enclosed within simple geometric shapes covered on some sides with a thin homogeneous cortical layer. |
| Bone Segment (BS) | A conventionally separated/distinguished segment of a bone containing AM. It is assumed that each segment has a homogeneous microstructure and cortical layer thickness. Segment sizes are selected so that they correspond to the average standard morphometric parameters of bone described in published papers or are easily deduced from them. |
| Basic (Bone) Phantom Segment (BPS) | Tissue-equivalent mathematical model that describes a bone segment as a combination of target and source tissue in voxel representation, with a simple geometric shape; modeled with population-averaged bone parameters. |
| Supplemental Phantom Segment (SPS) | Tissue-equivalent mathematical model that describes a bone segment as a combination of target and source tissue in voxel representation, with a simple geometric shape; modeled with parameters within the range of population variability; several supplemental phantoms represent the limits of some parametric variabilities. |

**Table 2. Main characteristics of bone microstructure.**

| Parameter | Abbreviation | Units |
|---|---|---|
| Bone volume | BV | mm³ |
| Tissue volume or total volume | TV | mm³ |
| Volume fraction of bone (trabeculae) in a tissue site | BV/TV | rel.units or % |
| Cortical thickness | Ct.Th | mm |
| Trabecular (rod) thickness | Tb.Th | mm |
| Trabecula separation (Inter-trabecular distance or space) | Tb.Sp | mm |

**Number of studied persons.** Each study should include the measurements of more than one subject; measurements of a single individual can be included in the absence of other data or if a set of measurements has been performed (e.g., set of ribs; set of vertebrae). A weighting factor ($W_N$) takes into account the number of subjects ($N$) included in a study: $W_N = 1$, if $N \geq 25$; $W_N = N/25$, if $N < 25$.

**Reference age.** Reference ages were established: newborn, 1-, 5-, 10-, 15-year-olds and adults (respectively, 0-Y, 1-Y, 5-Y, 10-Y, 15-Y, adult). For newborns, published data were used from measurements of late fetus, stillbirths and children up to three months; we mainly used the data for neonates and stillbirths. For a 1-year-old, data on children aged 0.75–2 years were used, for other ages, data on children whose age differs from the reference age by no more than 2 years were used.

For analysis of adult trabecular microstructures and cortical thickness, the data for persons aged in the range 20–60 years were used for men; and in the range 20–50 years for women; data for individuals characterized as "adults" were also included in analysis. The choice of age ranges is determined by the known changes in trabecular bone parameters due to age-related changes in sex-steroid hormone level. A weighting factor $W_A = 0.75$ is applied if individual data were not available and the upper limit of the age-range in the study group exceeded 50 for female and 59 for male, or when the age was indicated as "adults". The total weighting factor for microstructure parameters is the product $W_N \times W_A$. For analysis of adult bone sizes only $W_N$ was considered. It was considered that the bone dimensions during adulthood do not change significantly, in contrast to the trabecular bone characteristics.

**Gender/sex differences.** Bone parameters were assumed to be unnecessary for the 0-Y to 10-Y ages. For ages 15 years and older, the analysis of the published data was carried out to assess the effect of gender/sex.

## Considered methods for assessing bone parameters

*Method of trabecular bone measurements* used by authors in their studies: histomorphometry and micro-CT (computed tomography) data (pixel resolution < 40 μm). Measured bone parameters correspond to the generally accepted nomenclature [25,26] (Table 2). Studies describing the full set of spongiosa parameters were considered first.

*Method of cortical thickness measurements* is micro-CT and histomorphometry, if Ct.Th < 1 mm, as well as measurements with a micrometer, CT (if Ct.Th > 1 mm).

*Method of bone-size measurements* used by authors: postmortem measurements of individual bones using an anthropometric box and/or calipers of various designs (mechanical or electronic); measurements of images obtained in vivo by CT, MRI, as well as X-ray images; evaluation of 3D coordinates of reference points on the bone surface and calculation of the distance between points. Measured bone sizes are the standard parameters used in anthropometry and practical medicine (for example, the length and diameter of bone sections). In addition, published images provided with a scale (e.g., [20,28]) were analyzed. We also used the published estimates of bone-size proportions (e.g., [29,30]) to calculate some dimensional parameters.

If published data were unavailable for specific ages or bone segments, two educational collections of bones were used to assess the bone sizes of adults: an anatomical collection of bones of the South Ural State Humanitarian

Pedagogical University (collection SUSHPU) and the collection of the South Ural State Medical University (collection SUSMU). These measured bones had no evidence of bone damage or disease and were not subdivided into groups according to the sex. Measurements were performed by members of the Biophysics Laboratory (Sharagin P. and Shishkina E.) using a mechanical caliper.

### Considered methods to assess the duration of hematopoiesis in different parts of the skeleton

The FLT-PET method (positron-emission tomography with [18F] -Fluor-3-deoxy-3-L-fluorothymidine) is considered by us as the most accurate. FLT-PET data [31] were the basis for estimation of AM skeletal distribution in adults [6]. Unfortunately, there are no appropriate PET data for children. As humans age, the hematopoietic sites contain varying proportions of AM and IM. Minimal or no AM present in a bone segment would no longer be modeled. We used published data from MRI studies to determine whether AM or the fatty IM is in a bone segment of specific age [32–34], considering that a high-intensity signal is typical for fat (adipose tissue) and a low-intensity signal is typical for AM. The evaluated data revealed 3 or 4 levels of signal intensity used to determine the stage of conversion from AM to IM. Age boundaries of fatty degeneration of AM have a large individual variability. We assumed that AM was converted to IM if the MRI signal corresponds to the presence of greater than 80% adipose tissue in the examined persons; otherwise, the presence of AM is assumed.

### Statistical methods used

Bone parameters were assumed to be normally distributed. For dimensional parameters this is indicated by the authors of the studies (for example, [35]). For micro-parameters (trabecular thickness), authors indicated some asymmetry of distributions [36–38]; however, the normal distribution can also be applied for data description. If the published data on individual measurements were available, we combined them according to reference ages then calculated the arithmetic mean and standard deviation (SD). If the published data presented the results as mean values for different age periods, we averaged the results of several studies with the SD taken as an average value in a set of estimates. The range of minimum and maximum values was taken either in accordance with the range indicated by the authors or calculated as a range of two standard deviations from the mean.

### Analysis of the AM distribution in human bones for bone segments and reference ages

For modeling purposes, it is necessary to describe the microstructures and bone-sizes of hematopoietic sites to account for at least 90–95% of the total skeletal AM for each reference age.

In newborns, all bone marrow is considered as active; therefore, almost the entire set of modeled bone parts were included. In adults, hematopoiesis occurs at the following sites [13,31]: skull, vertebrae (cervical, thoracic, and lumbar), sacrum, pelvic bones, scapula, clavicle, ribs, sternum, proximal humerus, and proximal femur. In other words, by adulthood, hematopoiesis gradually ceases in the bones of the extremities except for the proximal part of the femur and humerus. For long tubular bones, the age-dynamics of the AM conversion into IM was considered in detail to determine the list of bone segments that should be modeled for each reference age. Epiphyses of long bones, facial bones, and vertebrae merited more detailed consideration. Attention was also paid to the scapula to determine at what age hematopoiesis ceases in its central (flat) part.

Table 3 summarizes the analysis of published data and assumed reference ages relevant for modeling of specific skeletal sites. As indicated in the final column, hematopoiesis quickly ceases and AM is replaced by adipose tissue in the shaft of tubular bones. The information from Table 3 drove our decision to exclude modeling of the epiphyses of the tubular bones because hematopoiesis ceases there early in life during mineralization. In 0-Y and 1-Y children, for whom epiphyseal hematopoiesis could be relevant, most of the epiphyses consist of cartilaginous

**Table 3. Published data and assumed reference ages important for modeling of specific bone segment.**

| Hematopoietic site | Published data description | SPSD-modeled reference ages |
|---|---|---|
| **Femur** | Shaft. According to [42] (n = 77) the conversion of AM begins at the ages of 1–5 years; at the ages of 6–10 years the replacement of AM with adipose tissue ends. Similar results were obtained by [43] (n = 52) and [44] (n = 381). Somewhat different results were obtained by [45] (n = 81), according to these data, the conversion of AM ends at the ages of 1–5 years. Epiphysis. According to [42] (n = 77) and [46] (n = 381) in children of the first year of life an MRI signal of different intensity is observed, which indicates the presence of a certain proportion of AM in epiphyses. Examinations of children ages 1–5 years [42–45] showed that the epiphyses at this age are filled mainly with adipose tissue. However, [44] observed the presence of AM (heterogeneous signal) in the proximal epiphysis in about 40% of children ages 1–3 years. Distal and proximal parts. According to the data of the above-listed authors, at the ages of 1–5 years, the metaphysis of the femur is filled with AM. At 6–10 years of age, the distal and proximal metaphysis is characterized by a heterogeneous signal, i.e., a gradual AM conversion occurs. According to most authors, after the age of 10 the distal metaphysis does not contain AM, however, [44] describes the presence of AM in the metaphysis in children aged 11–15 years in a small percentage of cases. | Shaft: 0-Y;1-Y;5-Y Epiphyses: not modeled Dist: 0-Y; 1-Y; 5-Y, 10-Y Prox: all ages |
| **Tibia and fibulae** | Taccone et al. [44] indicates that the conversion of AM to adipose tissue in the diaphysis occurs at the ages of 6–10 years. According to the same authors, at the ages of 6–10 years, the AM conversion ends in both metaphysis of the fibula and the distal metaphysis of the tibia. However, in the proximal metaphysis of tibia, the conversion ends later, at the ages of 11–15 years. These data are not entirely consistent with the histological studies of [46], who observed AM in the proximal metaphysis of the tibia only for children under the age of 10 years. As in the femur, in the epiphyses of these bones, MRI signals of varying intensity are detected during the first year of life [44]; for older ages, AM in the epiphyses was not detected. | Shaft: 0-Y;1-Y; 5-Y Prox: 0-Y; 1-Y; 5-Y; 10-Y Dist: 0-Y;1-Y; 5-Y |
| **Humeri** | According to [47] (n = 91), in diaphysis, AM conversion to adipose tissue begins at 1 year and practically ends by the age of 5 years. Similar results were described in [44]. In the distal metaphysis, conversion occurs later, at the ages of 11–15 years, when the authors either do not find AM at all or find it in a small number of children [44,47] in the proximal metaphysis, hematopoiesis continues in adults. In the distal epiphysis, the conversion is completed by the age of one year; in the proximal epiphysis, AM is observed for a long period of time, up to the age of 16 years [44], after that, heterogeneous signals (presence of AM) are observed in rare cases. | Shaft: 0-Y;1-Y; 5-Y Prox: all ages Dist: 0-Y;1-Y |
| **Radial, ulna bones** | According to [44], in radial bone (diaphysis, distal epiphysis and distal and proximal metaphysis) AM is replaced by adipose tissue by the age of 5. The same age pattern was assumed for radial proximal epiphysis and all parts of ulna. | Shaft: 0-Y;1-Y Prox: 0-Y;1-Y Dist: 0-Y;1-Y |
| **Phalanges of fingers, bones of wrist and tarsus, ankle, patella** | According to the published data [16,18,44], AM in these bones is replaced by adipose tissue by the end of the first year of life. | Whole bone: 0-Y |
| **Scapula** | At the ages of 1–3 years, the AM is converted into adipose tissue in the central (flat) part of the scapula [44]. In the remaining segments of the scapula (acromion, lateral margin and glenoid part), active hematopoiesis continues throughout life. | Body: 0-Y; 1-Y Other parts: all ages |
| **Facial skull** | Yamada et al. (n = 45) [48] provided a detailed description of fatty changes in mandible: the gradual conversion begins in the body and angle region at the ages of 2–5 years. As a result, by the age of 20, AM is observed in mandibular ramus, angle and body in 60%, 65% and 30% of persons, respectively. According to [49] (n = 324), AM is mainly replaced by adipose tissue in the mandible at the age of 8.5 years. In the upper jaw (maxilla, zygomatic bone), AM is not observed after the age of 3 months. | Not modeled |

Notes: *Prox*- proximal metaphysis; *Dist* -distal metaphysis; *Shaft or tube* – the central part of the bone (syn. diaphysis). In a newborn, shaft filled with spongiosa contains AM, which changes with age to IM without trabecular structures; *Distal and proximal distal parts* – extended or "cone-shaped" areas of bone (syn. metaphysis); small base of "truncated cone" is adjacent to the shaft, large base – to the growth zone (epiphyseal line); AM in them is gradually converted into IM. Note that in children, shaft and distal and proximal parts are combined in measurements of linear sizes, and thus, «maximal diaphysis length» includes shaft and distal and proximal ends [50,51]; *Epiphyses* are the most extreme areas of the bone (for example, head of humerus and femur), formed from secondary centers of ossification; in children they are separated from the metaphysis by a cartilaginous plate (growth zone or epiphyseal line); mineralized epiphyses filled with spongiosa, where AM is rapidly converted into IM.

tissue within which there are small centers of ossification. The facial bones of the skull were not modeled for any age due to the small proportion of AM in these bones relative to the entire skeleton. In newborns and infants, the mass proportion of facial bones in the skull is about 13% and a significant part of the mandibular and maxilla body is occupied by developing teeth. Although the relative proportion of the facial bones to the overall skull increases with age, there is an age-related decrease in active hematopoiesis in the facial region. AM is recorded in adults only in small areas (ramus mandibulae and angle). We did not model the **s**mall ossification centers in the processes of the vertebrae in 0-Y nor 1-Y ages. For the remaining ages, hematopoiesis in the vertebra processes was modeled, except for vertebra pedicles, which make up only about 3% of the vertebra volume and contain about 1% of vertebra AM. Similarly, we did not model the coccygeal vertebrae and the sacral vertebrae [39–41]. The analysis of radiographs of the sternal centers of ossification [20] showed that these centers which are located within the cartilaginous sternum are small in 0-Y, 1-Y ages, and vary widely in number and shape, so it was decided not to model them.

Tables 4–6 present the modeled hematopoietic skeletal sites and bone parts showing the ages that the cessation of active hematopoiesis is assigned in the model (indicated by darkened fields in the tables). Tables 4–6 also describe the parameters of the trabecular bone (BV/TV, Tb.Th, and Tb.Sp) for person aged 0–60 for men and 0–50 for women. Presented parameters of microstructure (micro-parameters) are discussed in detail in the next section.

**Table 4. Ratio BV/TV assumed in SPSD model (based on published data S1 Femur–S14 Hand Foot).**

| Skeletal site | Bone part | 0-Y | | 1-Y | | 5-Y | | 10-Y | | 15-Y | | Adult | |
|---|---|---|---|---|---|---|---|---|---|---|---|---|---|
| | | BV/TV | SD | BV/TV | SD | BV/TV | SD | BV/TV | SD | BV/TV | SD | BV/TV | SD |
| Clavicle | Body | 0.29 | 0.09 | 0.29 | 0.09 | 0.15 | 0.03 | 0.15 | 0.03 | 0.15 | 0.03 | 0.15 | 0.03 |
| | Ends | 0.29 | 0.09 | 0.29 | 0.09 | 0.29 | 0.09 | 0.29 | 0.09 | 0.29 | 0.09 | 0.29 | 0.09 |
| Femur | Proximal, all segments | 0.37 | 0.16 | 0.22 | 0.07 | 0.35 | 0.06 | 0.35 | 0.06 | 0.35 | 0.06 | 0.11-0.17 | 0.04-0.06 |
| | Diaphysis | 0.37 | 0.11 | 0.22 | 0.07 | 0.26 | 0.06 | | | | | | |
| | Distal, all segments | 0.37 | 0.11 | 0.22 | 0.07 | 0.26 | 0.06 | 0.26 | 0.06 | | | | |
| Fibula and tibia | Diaphysis | 0.35 | 0.07 | 0.20 | 0.04 | 0.25 | 0.04 | | | | | | |
| | Distal and proximal | 0.35 | 0.07 | 0.20 | 0.04 | 0.25 | 0.04 | 0.25 | 0.04 | | | | |
| Humerus | Proximal, all segments | 0.28 | 0.07 | 0.22 | 0.07 | 0.22 | 0.07 | 0.22 | 0.07 | 0.22 | 0.07 | 0.06 | 0.02 |
| | Diaphysis | 0.28 | 0.07 | 0.22 | 0.07 | 0.22 | 0.07 | | | | | | |
| | Distal, all segments | 0.28 | 0.07 | 0.22 | 0.07 | 0.22 | 0.07 | 0.22 | 0.07 | | | | |
| Pelvis | Iliac segments | 0.32 | 0.07 | 0.23 | 0.03 | 0.25 | 0.02 | 0.25 | 0.02 | 0.25 | 0.02 | 0.19 | 0.05 |
| | Ischium and Pubic | 0.32 | 0.07 | 0.23 | 0.03 | 0.25 | 0.02 | 0.25 | 0.02 | 0.25 | 0.02 | 0.17-0.25 | 0.01-0.03 |
| Radii+Ulnae | All segments | 0.21 | 0.05 | 0.16 | 0.05 | 0.16 | 0.05 | | | | | | |
| Ribs | All segments | 0.19 | 0.05 | 0.29 | 0.10 | 0.20 | 0.06 | 0.20 | 0.06 | 0.12 | 0.04 | 0.12 | 0.04 |
| Scapula | All segments | 0.28 | 0.08 | 0.22 | 0.08 | 0.22 | 0.08 | 0.22 | 0.08 | 0.22 | 0.08 | 0.22 | 0.08 |
| Skull | Flat bones (vault) | 0.52 | 0.12 | 0.52 | 0.12 | 0.52 | 0.12 | 0.52 | 0.12 | 0.52 | 0.12 | 0.52 | 0.12 |
| Sternum* | All segments | | | | | 0.15 | 0.04 | 0.15 | 0.04 | 0.15 | 0.04 | 0.15 | 0.04 |
| Vertebrae-C | All segments | 0.60 | 0.07 | 0.20 | 0.04 | 0.21 | 0.05 | 0.21 | 0.05 | 0.21 | 0.05 | 0.21 | 0.05 |
| Vertebrae-L+T+Sacrum | All segments | 0.45 | 0.15 | 0.14 | 0.04 | 0.14 | 0.04 | 0.14 | 0.04 | 0.14 | 0.04 | 0.15-0.16 | 0.03-0.05 |
| Wrist, ankle, foot and hand | All segments | 0.22 | 0.14 | | | | | | | | | | |

Note: dark fields mean that bone segments were not modeled due to the absence (or minimal amount) of active hematopoiesis. L – lumbar; T – thoracic; C – cervical.

* Small centers of sternum ossification in 0-Y and 1-Y are not modeled.

Table 5. Trabecular thickness values (Tb.Th, mm) assumed in SPSD model (based on published data S1 Femur–S14 Hand Foot).

| Skeletal site | Bone part | 0-Y | | 1-Y | | 5-Y | | 10-Y | | 15-Y | | Adult | |
|---|---|---|---|---|---|---|---|---|---|---|---|---|---|
| | | Tb.Th | SD | Tb.Th | SD | Tb.Th | SD | Tb.Th | SD | Tb.Th | SD | Tb.Th | SD |
| Clavicle | Body | 0.15 | 0.02 | 0.15 | 0.02 | 0.19 | 0.06 | 0.19 | 0.06 | 0.19 | 0.06 | 0.19 | 0.06 |
| | Ends | 0.15 | 0.02 | 0.15 | 0.02 | 0.15 | 0.02 | 0.15 | 0.02 | 0.15 | 0.02 | 0.14 | 0.02 |
| Femur | Proximal, all segments | 0.11 | 0.017 | 0.16 | 0.06 | 0.24 | 0.053 | 0.24 | 0.053 | 0.24 | 0.053 | 0.14−0.19 | 0.02-0.03 |
| | Diaphysis | 0.11 | 0.017 | 0.16 | 0.06 | 0.24 | 0.053 | | | | | | |
| | Distal, all segments | 0.11 | 0.017 | 0.16 | 0.06 | 0.24 | 0.053 | 0.24 | 0.053 | | | | |
| Fibula and tibia | Diaphysis | 0.075 | 0.007 | 0.089 | 0.008 | 0.126 | 0.017 | | | | | | |
| | Distal and proximal | 0.075 | 0.007 | 0.089 | 0.008 | 0.126 | 0.017 | 0.212 | 0.028 | | | | |
| Humerus | Proximal, all segments | 0.102 | 0.052 | 0.174 | 0.033 | 0.208 | 0.028 | 0.208 | 0.028 | 0.208 | 0.028 | 0.1 | 0.018 |
| | Diaphysis | 0.102 | 0.052 | 0.174 | 0.033 | 0.208 | 0.028 | | | | | | |
| | Distal, all segments | 0.102 | 0.052 | 0.174 | 0.033 | 0.208 | 0.028 | 0.208 | 0.028 | | | | |
| Pelvis | Iliac segments | 0.166 | 0.024 | 0.123 | 0.024 | 0.153 | 0.016 | 0.155 | 0.016 | 0.155 | 0.016 | 0.13 | 0.02 |
| | Ischium and Pubic | 0.166 | 0.024 | 0.123 | 0.024 | 0.153 | 0.016 | 0.155 | 0.016 | 0.155 | 0.016 | 0.17-0.30 | 0.03-0.02 |
| Radii+Ulnae | All segments | 0.08 | 0.02 | 0.134 | 0.02 | 0.16 | 0.02 | | | | | | |
| Ribs | All segments | 0.136 | 0.048 | 0.231 | 0.078 | 0.231 | 0.078 | 0.231 | 0.078 | 0.147 | 0.018 | 0.147 | 0.018 |
| Scapula | All segments | 0.12 | 0.1 | 0.192 | 0.1 | 0.24 | 0.1 | 0.24 | 0.1 | 0.24 | 0.1 | 0.24 | 0.1 |
| Skull | Flat bones (vault) | 0.29 | 0.09 | 0.29 | 0.09 | 0.29 | 0.09 | 0.29 | 0.09 | 0.29 | 0.09 | 0.29 | 0.09 |
| Sternum | All segments | | | | | 0.135 | 0.04 | 0.15 | 0.05 | 0.15 | 0.05 | 0.15 | 0.05 |
| Vertebrae-C | All segments | 0.25 | 0.014 | 0.18 | 0.024 | 0.14 | 0.02 | 0.14 | 0.02 | 0.14 | 0.02 | 0.14 | 0.02 |
| Vertebrae-L+T+Sacrum | All segments | 0.096 | 0.04 | 0.096 | 0.04 | 0.096 | 0.04 | 0.118 | 0.02 | 0.118 | 0.02 | 0.1-0.15 | 0.013-0.03 |
| Wrist, ankle, foot and hand | All segments | 0.12 | 0.032 | | | | | | | | | | |

Note: dark fields mean that bone segments were not modeled due to the absence (or minimal amount) of active hematopoiesis. L – lumbar; T – thoracic; C – cervical.

∗ Small centers of sternum ossification in 0-Y and 1-Y are not modeled.

## Parameters for trabecular bone microstructure (micro-parameters) in the hematopoietic sites of human skeleton

### Sex differences in bone microstructure and their potential significance

Sex differences in skeletal development are subtle during childhood. This applies to both dimensional parameters and mineral metabolism rates. In this regard, parameters of the biokinetic model for bone-seeking radionuclides are also not age-dependent up to the age of 10 years [52]. In contrast, the most pronounced effect of sex on bone microstructure can be expected in adults. We examined this observation in relation to adults under age 50 and our reviews have shown the following. Stauber [53] found no sex differences in the parameters of the bone-microstructure (e.g., BV/TV) for two bone parts (femur head and lumbar vertebra), but found a significant dependence on age (young adults vs. elderly) and bone

Table 6. Trabecular separation values (Tb.Sp, mm) used for calculation in SPSD model (based on published data S1 Femur–S14 Hand Foot).

| Skeletal site | Bone part | 0-Y | | 1-Y | | 5-Y | | 10-Y | | 15-Y | | Adult | |
|---|---|---|---|---|---|---|---|---|---|---|---|---|---|
| | | Tb.Sp | SD | Tb.Sp | SD | Tb.Sp | SD | Tb.Sp | SD | Tb.Sp | SD | Tb.Sp | SD |
| Clavicle | Body | 1.1 | 0.14 | 1.1 | 0.14 | 1.1 | 0.14 | 1.1 | 0.14 | 1.1 | 0.14 | 1.1 | 0.14 |
| | Ends | 1.1 | 0.14 | 1.1 | 0.14 | 1.1 | 0.14 | 1.1 | 0.14 | 1.1 | 0.14 | 1.1 | 0.14 |
| Femur | Proximal, all segments | 0.388 | 0.103 | 0.538 | 0.11 | 0.538 | 0.077 | 0.538 | 0.077 | 0.538 | 0.077 | 0.78-0.99 | 0.1-0.2 |
| | Diaphysis | 0.388 | 0.103 | 0.538 | 0.11 | 0.538 | 0.077 | | | | | | |
| | Distal part, all segments | 0.388 | 0.103 | 0.538 | 0.11 | 0.538 | 0.077 | 0.538 | 0.077 | | | | |
| Fibula and tibia | Diaphysis | 0.49 | 0.11 | 0.735 | 0.084 | 0.735 | 0.084 | | | | | | |
| | Distal and proximal | 0.49 | 0.11 | 0.735 | 0.084 | 0.735 | 0.084 | 0.735 | 0.084 | | | | |
| Humerus | Proximal, all segments | 0.36 | 0.041 | 0.58 | 0.27 | 0.58 | 0.188 | 0.58 | 0.188 | 0.58 | 0.188 | 2.37 | 0.6 |
| | Diaphysis | 0.36 | 0.041 | 0.58 | 0.27 | 0.58 | 0.188 | | | | | | |
| | Distal, all segments | 0.36 | 0.041 | 0.58 | 0.27 | 0.58 | 0.188 | 0.58 | 0.188 | | | | |
| Pelvis | Iliac segments | 0.32 | 0.1 | 0.481 | 0.112 | 0.481 | 0.112 | 0.459 | 0.07 | 0.459 | 0.07 | 0.6 | 0.12 |
| | Ischium and Pubic | 0.32 | 0.1 | 0.481 | 0.112 | 0.481 | 0.112 | 0.6 | 0.07 | 0.750 | 0.07 | 0.75-1.0 | 0.15-0.4 |
| Radii+Ulnae | All segments | 0.51 | 0.121 | 0.765 | 0.121 | 0.765 | 0.11 | | | | | | |
| Ribs | All segments | 0.52 | 0.051 | 0.51 | 0.071 | 0.51 | 0.071 | 0.51 | 0.071 | 0.82 | 0.1 | 0.82 | 0.1 |
| Scapula | All segments | 0.482 | 0.243 | 0.964 | 0.221 | 0.964 | 0.221 | 0.964 | 0.221 | 0.964 | 0.221 | 0.96 | 0.221 |
| Skull | Flat bones (vault) | 0.57 | 0.2 | 0.57 | 0.2 | 0.57 | 0.2 | 0.57 | 0.2 | 0.57 | 0.2 | 0.57 | 0.2 |
| Sternum* | All segments | | | | | 1.0 | 0.06 | 1.0 | 0.06 | 1.0 | 0.06 | 1.0 | 0.06 |
| Vertebrae-C | All segments | 0.6 | 0.12 | 0.6 | 0.12 | 0.6 | 0.12 | 0.65 | 0.154 | 0.65 | 0.154 | 0.6 | 0.09 |
| Vertebrae L+T+Sacrum | All segments | 0.6 | 0.12 | 0.6 | 0.12 | 0.6 | 0.12 | 0.65 | 0.154 | 0.65 | 0.154 | 0.6 | 0.09 |
| Wrist, ankle, foot and hand | All segments | 0.25 | 0.13 | | | | | | | | | | |

Note: dark fields mean that bone segments were not modeled due to the absence (or minimal amount) of active hematopoiesis. L – lumbar; T – thoracic; C – cervical.

* Small centers of sternum ossification in 0-Y and 1-Y are not modeled.

site. Stauber [53] also points to the unreliable contribution of sex as a covariant for global morphometry across three sites (femoral head, lumbar vertebra, iliac crest). Several authors also note the absence of sex differences or indicated small differences (a few percent) for the femoral head [54–56], iliac bone [54,57,58], and vertebrae [59,60]. Thus, it was concluded that sex does not significantly vary the micro-parameters, and data for men and women of the same age were combined for parameter determination and modeling.

## Parameters of trabecular bone microstructure in specific bones

Our summary of published data on trabecular bone microstructure is presented in Tables 4–6. More detailed information on the bone microstructures data is provided in S1 Femur–S14 Hand Foot. We distinguish 3 groups of bone sites/segments according to data available.

Group 1 (proximal femur [S1 Femur], proximal humerus [S2 Humeri], lumbar and thoracic vertebrae [S6 Vertebra], ilium [S3 Pelvis], tibia [S5 Tibia], rib [S4 Rib]). Data sets (S1 Femur–S6 Vertebra) were available for adults, infants and most pre-adult age groups. For Group 1, the microstructure parameters were reliably determined from the measured data. Figs 2–4 illustrate the measured and approximated data of bone microstructures (BV, Tb.Th, and Tb.Sp, respectively) used in the SPSD model as summarized previously in Tables 4–6. Upon review of the data sets, the parameter values estimated for lumbar and thoracic vertebrae were accepted for *sacrum* (S7 Sacrum). The trabecular structure of the *fibula* (S8 Fibula) is assumed to be the same as that of the *tibia* (S5 Tibia).

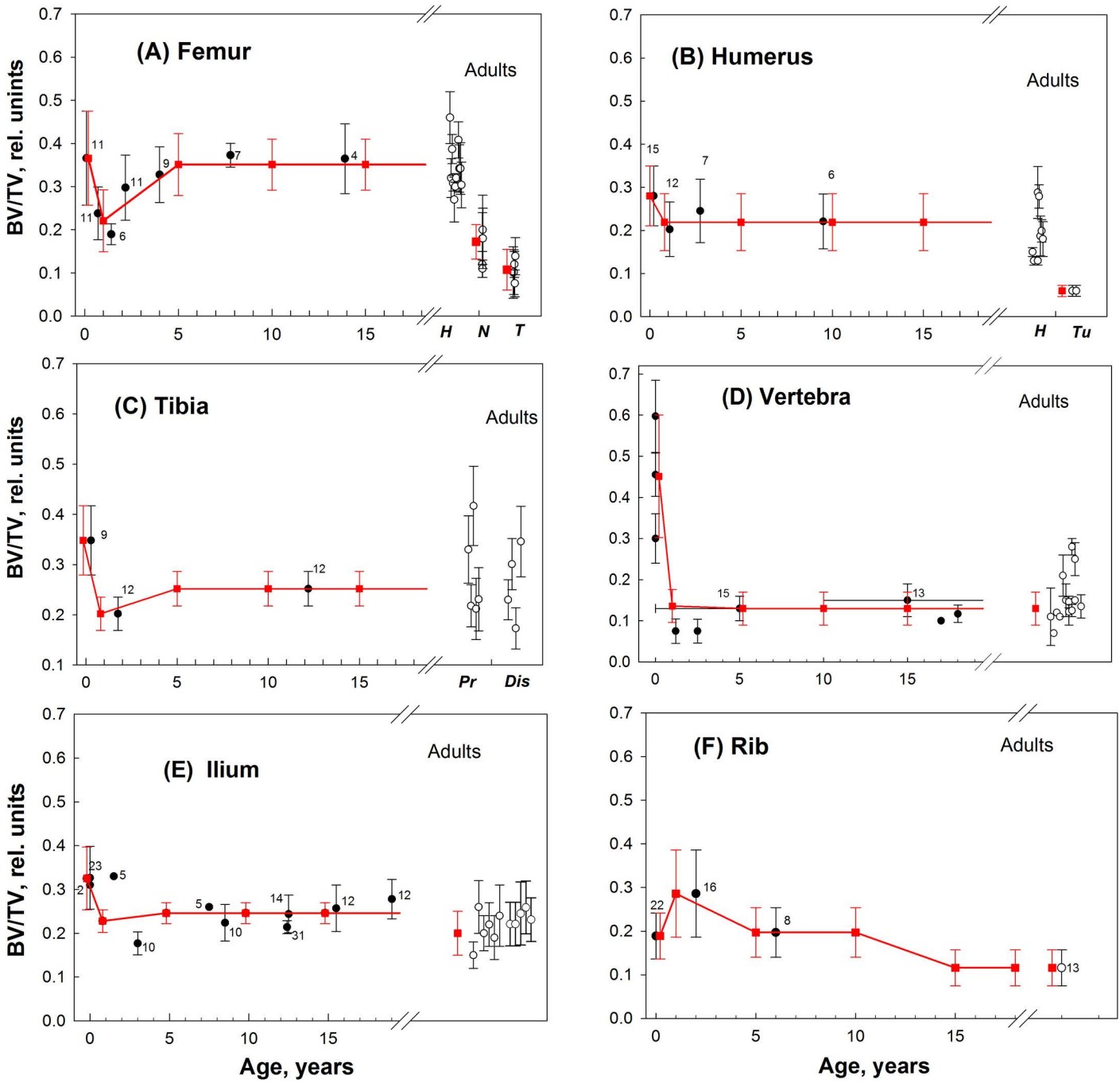

**Fig 2. Age dependences of BV/TV (in relative units) in different hematopoietic sites.** Black circles – group-average data for pre-adults; open circles – group-average data for adults (detailed author's data are given in Supporting Information S1 Femur–S6 Vertebra); the number near the circles indicates the number of persons; vertical bars represent the scatter (SD) of value in the groups; red squares and bars – values assumed for SPSD model. **A** – measured data and assumed model-values for proximal femur of pre-adults [15,61–65] and adults (total number of adult-persons measured (n = 214) separately for head (H) [56,64,66–69], neck (N) [70,71], and trochanter area (T) [70–74]. **B** – measured data and assumed model-values for proximal humerus of pre-adults [63,65,75] and adults (n = 69) separately for head (H) [69,76–78] and tuberculum area (Tu) [76]. **C**- measured data and assumed model-values for proximal tibia of pre-adults [79,80] and adults [81] separately for proximal (Pr) and distal (Dis) parts. **D**- the measured data and assumed model-values for thoracic and lumbar-vertebra-body for pre-adults [63,82–84] and adults (n = 155) [59,82,83,85–92]. **E**- mainly shows the data on in vivo biopsy of iliac crest from pre-adults [93–96] and adults (n = 347) [57,58,88,97–103] and assumed model-values. **F**- assumed model-values and values measured along the long axis of 5-7 ribs for pre-adult [104] and data [105] for adults.

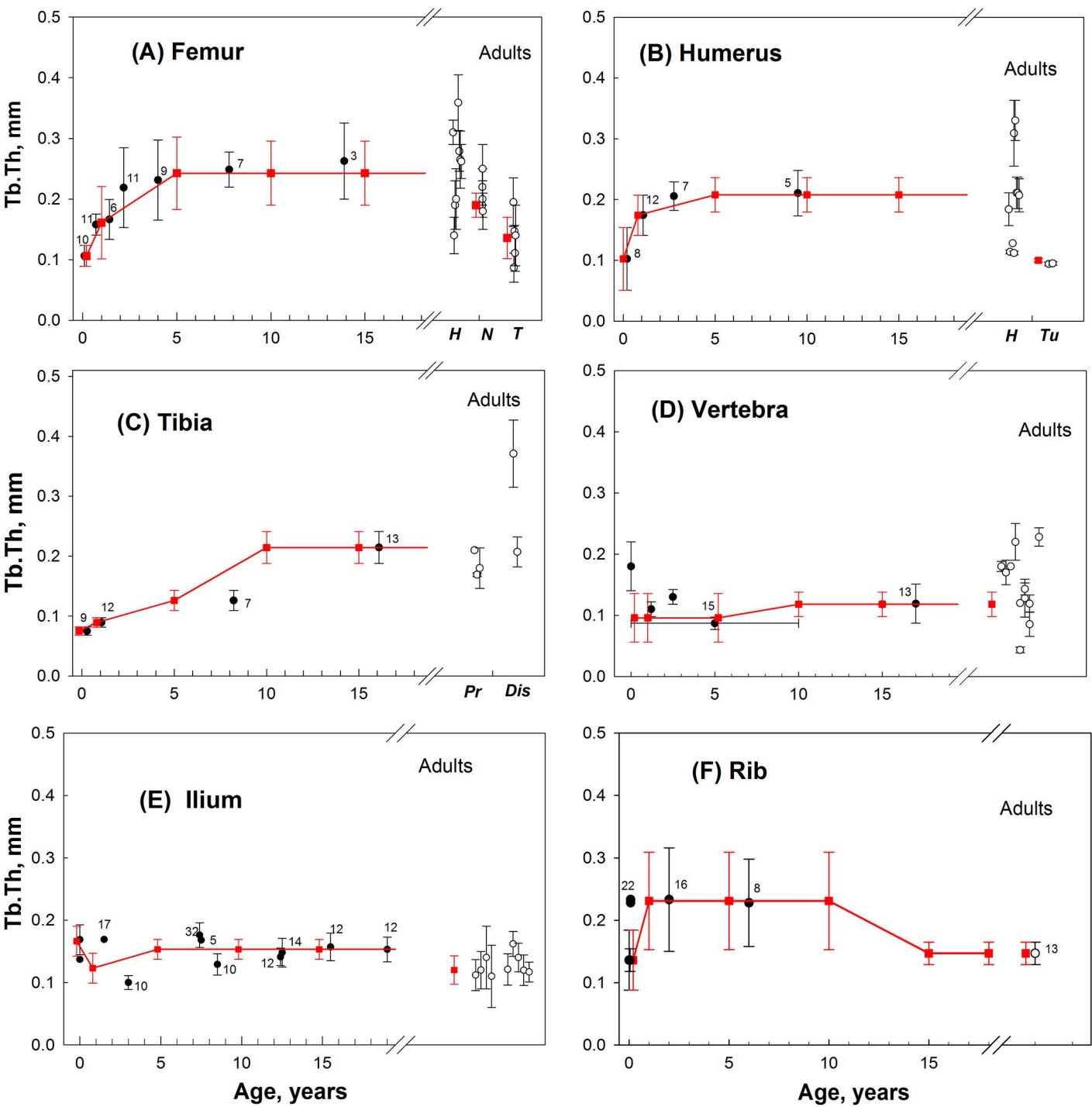

**Fig 3. Age dependences of Tb.Th (mm) in different hematopoietic sites.** Black circles – group-average data for pre-adults; open circles – group-average data for adults (detailed author's data are given in Supporting Information S1 Femur–S6 Vertebra); the number near the circles indicates the number of persons; vertical bars represent the scatter (SD) of value in the groups; red squares and bars – values assumed for SPSD model. **A** – measured data and assumed model-values for proximal femur of pre-adults [61–65] and adults (total number of adult-persons measured n = 214) separately for head (*H*) [56,68,69,72], neck (*N*) [70,71], and trochanter area (*T*) [70–74]. **B** – measured data and assumed model-values for proximal humerus of pre-adults [63,65,75,79] and adults (n = 69) separately for head (*H*) [69,76–78] and -tuberculum area (*Tu*) [76]. **C** – measured data and assumed model-values for proximal tibia of pre-adults [79,80] and adults separately for proximal (Pr) n = 90 [80,81] and distal (Dis) parts [80]. **D** – measured data for

thoracic and lumbar vertebra body for pre-adults [63,82–84] and adults (n = 155) [59,68,82,83,87,88,91,92,106–109]. **E** – measured data and assumed model-values on in vivo biopsy of iliac crest from pre-adults [93–96,110] and adults (n = 347) [58,88,99,101]. **F**- assumed model-values and values measured along the long axis of 5-7 ribs for pre-adult [104] and data [105] for adults.

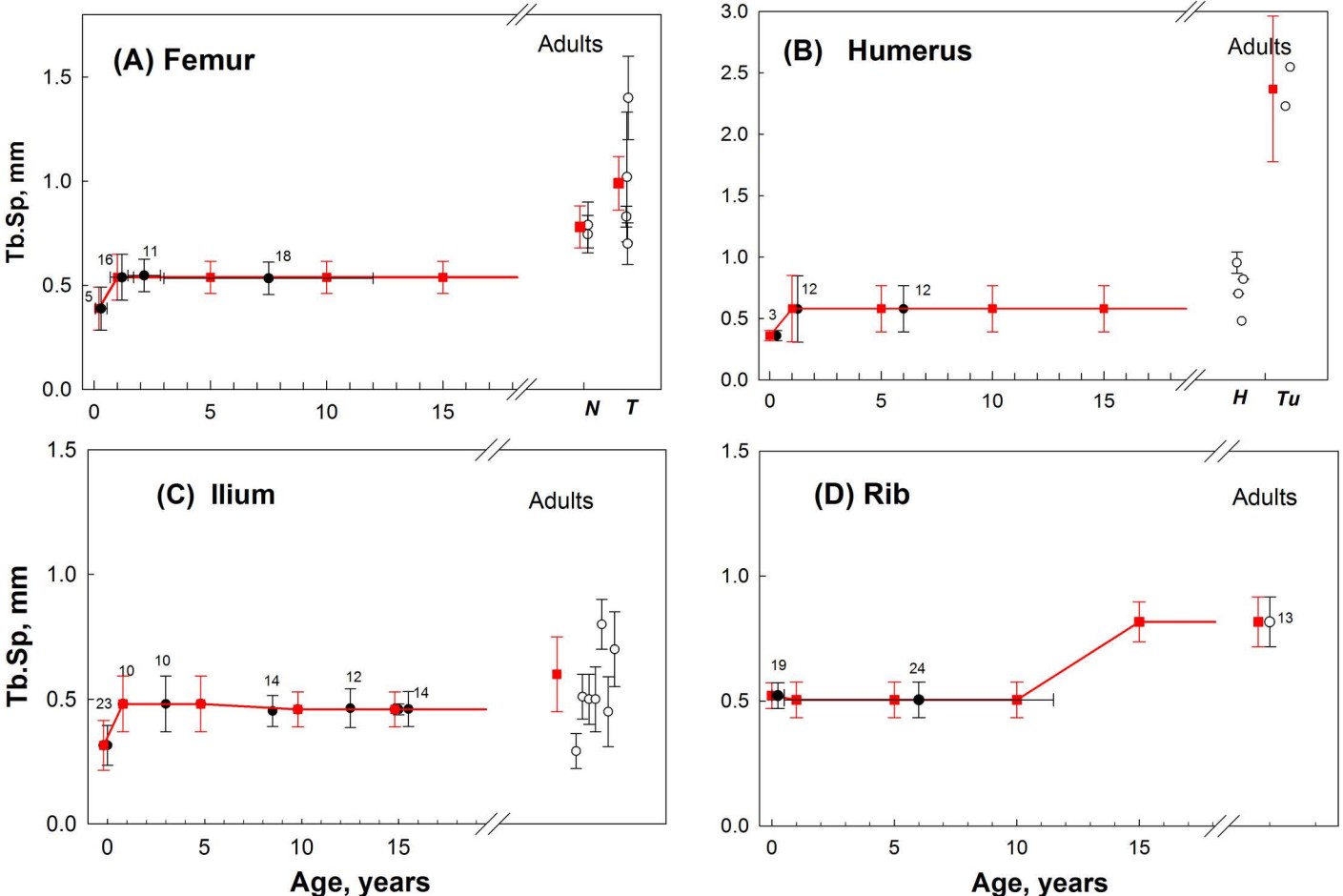

**Fig 4. Age dependences of Tb.Sp (mm) in different hematopoietic sites.** Black circles – group-average data for pre-adults; open circles – group-average data for adults (detailed author's data are given in Supporting Information S1 Femur–S4 Rib); the number near the circles indicates the number of persons; vertical bars represent the scatter (SD) of value in the groups; red squares and bars – values assumed for SPSD model. **A**- measured data and assumed model-values for proximal femur of pre-adults [61–65] and adults separately for neck (*N*), n = 37 [70,71], and trochanter area (*T*); n = 76 [70–73]. **B**- measured data and assumed model-values for proximal humerus of pre-adults [65,75,79] and adults separately for head (*H*) (n = 39) [76–78] and -tuberculum area (*Tu*) n = 30 [76]; we do not model the humeral head for any age; adult head data is shown for comparison. **C**- assumed model-values and data on in vivo biopsy of iliac crest from pre-adults [93–96] and adults (n = 285) [58,88,97,99–102]. **D** – assumed model-values and values measured along the long axis of 5-7 ribs for pre-adult [104] and data [105] for adult, n = 13.

As seen in Fig 2, for each Group 1 bone, maximum BV/TV values are observed in newborns (excluding rib). The age dependences for these bones are qualitatively similar. We observe a decrease in BV/TV by the age of 1-Y, which is expressed to varying degrees. After that, BV/TV values either increase (femur, ilium, and tibia) or remain at the same level (humerus, vertebrae). At the age of 15-Y, BV/TV values can be close to those for adults (vertebrae, femur and humerus heads), or significantly exceed the adult values (metaphyseal parts of the humerus and femur, i.e., femur, neck and trochanter area and humerus tuberculum area). Rib data vary from the described regularity and has the fewest measured

data available of all Group 1 bones. The maximum BV/TV values for rib are observed in 1-Y, rather than in newborns. For Group 1 Tb.Th values (Fig 3), the minimum values are observed in newborns (except for the vertebra), followed by a gradual increase in Tb.Th. At the age of 15-Y, Tb.Th values can be close to those for adults (vertebrae, ilium) or significantly exceed the values for adults. As for Group 1 Tb.Sp values (Fig 4) fewer measured data are available. To illustrate, data are given for four sites, which show very similar dynamics of age-related changes in Tb.Sp: minimum values are observed in newborns; by the age of 1 year, the values increase in all sites studied (except for the rib). As the analysis of the sites with the most complete data (ilium, femur) shows, the Tb.Sp values do not change until the ages of 12–15 years. However, Tb.Sp is significantly higher in adults than in pre-adults.

Figs 2–4 present microparameter data for the proximal part of tubular bones (femur, humerus, tibia). The evaluation of micro-parameters for the distal parts requires a more detailed description. Data for the distal part of *femur* (S1 Femur) and *tibia* (S5 Tibia) satisfying the analysis conditions were found only for adults. Saers et al. [81] presents a comparison of micro-parameters of the distal and proximal femur and tibia in three human populations. *Femur:* In all cases, within the same population, significant differences were found between BV/TV of femoral distal and proximal ends. Differences amounted to 25–30% and were revealed both in data averaged over three populations and within individual populations. For people 5 years of age and older BV/TV of the distal part was assumed to be 25% lower than in the proximal part. However, there were no significant differences in Tb.Th. Thus, similar Tb.Th values are accepted both for the distal and proximal parts. It is assumed that the variability of BV/TV and Tb.Th (SD value) of the distal part is equal to that of the proximal part. Tb.Sp was not estimated in [81], and Tb.Sp was taken to be the same as for the proximal part. *Tibia*: a comparison of micro-parameters of the distal and proximal parts [81] has shown no significant differences between the distal and proximal parts in terms of BV/TV and Tb.Th. Thus, for the distal part, the same parameters (including Tb.Sp) are accepted as for the proximal one (Tables 4–6). For *humerus*, we did not find the parameters for the distal part, thus, they were taken to be the same as for the proximal part.

Figs 2–4 present the sets of measured data for *ilium* (S3 Pelvis). However, the hematopoietic region of the pelvic bone also includes other bone parts, i.e., ischium and pubic bones; they represent separate bones (separate centers of ossification) up to the ages of 12–15. For ischium and pubis (all segments, all ages), BV/TV was taken to be the same as for the ilium. Similarly, Tb.Th and Tb.Sp for 0, 1, 5 year ages was taken to be the same as for the ilium. For older ages, a gradual Tb.Th and Tb.Sp increase for the ischium and pubic bones was assumed so as to reach an adult-value by the age of 25 (adult value according to [111,112]). SD for pubis and ischium was taken as equal to that for ilium.

**Group 2 (skull, cervical vertebrae, radius).** For the skull, cervical vertebrae, and radius, quantitative data for infants and adults ("extreme" age points) and qualitative data reflecting age dynamics (i.e., images of histological sections or high-resolution peripheral quantitative computed tomography [HR-pQCT]) were available. Note that the published data on Tb.Th from HR-pQCT (pixel resolution 82 μm) studies significantly differs from μCT data (resolution <40 μm) and cannot be used for SPSD parameter estimations.

*Skull* (S9 Skull). Rodriguez-Florez [113] estimate the value BV/TV in infants (parietal bones) as equal to $0.5 \pm 0.1$ (n = 18). A similar BV/TV value was derived from published data for adults ($0.52 \pm 0.1$), (data averaged over several skull bones, n = 174). Image analysis of histological-sections of pre-adults [114] showed that the value of Tb.Th and Tb.Sp with its variability does not differ significantly from that for adults. Trabecular-bone-parameters were accepted as independent of age (Tables 4–6).

*Cervical vertebra* (S6 Vertebra) parameters for 0-Y and 1-Y were taken according to published data [63,84], though limited by the total number of measured persons (n = 4), for each person 5–7 vertebrae were measured. For children of 5-Y to 15-Y the adult-parameter values were accepted for BV/TV and Tb.Th. For cervical vertebra Tb.Sp values, lumbar and thoracic vertebra values were assigned. All accepted parameters are shown in Tables 4–6.

*Radius (distal part)* (S10 Radius Ulna). Parameter values for adults were derived from μCT data. Analysis of HR-pQCT measurements have shown, that values of BV/TV and Tb.Th are not age-dependent after the age of 5 years [90,115–123].

For children under 5 years, we assume that the dynamics of changes in micro-parameters is like that for the humerus. This means that in newborns and 1 year olds, BV/TV is 1.3 times higher than in children ≥ 5. In turn, Tb.Th of newborns is 2 times lower than in children ≥5 years; Tb.Th in one-year-old children are 20% lower than in children ≥ 5. The accepted values of the micro-parameters for radii are presented in Tables 4–6.

**Group 3 (scapula; sternum; clavicle; wrist, ankle, foot and hand).** This group includes bones for which adult-only data are available (scapula, sacrum, clavicle, sternum) or age-appropriate measurements are available only for adjacent bones (wrists, metacarpus, tarsus, metatarsus, phalanges of fingers and toes).

*Sternum* (S11 Sternum). Note that in 0-Y and 1-Y ages, the centers of ossification are very small, the sternum is mainly composed of cartilage, thus, the sternum of newborns is not included in the SPSD model. The age-dynamics for sternum was assigned like that of the vertebra. BV/TV corresponds to the adult level in the remaining age groups (5–15 years). The Tb.Th value reached adult levels by age 10-Y. For the age of 5-Y, the value was taken to be 10% lower than for adults (Tables 4–6).

*Scapula* (S12 Scapula). We assume that the dynamics of changes in the parameters of the scapula microstructure corresponds to that for the proximal humerus that forms one joint with the scapula. Thus, it is assumed that the values of BV/TV, Tb.Sp and Tb.Th for children of 5 years and older are equal to those for adults. In newborns, BV/TV is 1.3 times higher, and Tb.Th is 2 times lower than in adults; in 1-year-olds Tb.Th is 20% lower than in adults. The assigned values of the parameters for scapula are presented in Tables 4–6.

*Clavicle* (S13 Clavicle). The following dynamics of parameters are taken for 0-Y and 1-Y, the parameters for ends and body are the same and correspond to those for adult ends. For children older than 5-Y, all parameters correspond to the values for adults.

In the bones of the wrist, metacarpus, tarsus, metatarsus, phalanges of the fingers and toes (S14 Hand Foot), active hematopoiesis ends by the age of 1 year. Considering the comparative study [124], for children under the age of 1 year, it was assumed that all these bones have the same parameters of trabecular microstructure, which correspond to the values for a well-studied calcaneus [125] (Tables 4–6).

### Intra-specimen variability of trabecula microstructure in different bones

Intra-specimen variability reflects the changes in the parameters of the trabeculae (Tb.Th, Tb.Sp) within specific bones of a particular person. Values of intra-bone variability are used to determine the range of multiple 3-D grid deformations of the trabecular grid in the SPSD model (Fig 1d): changing the positions of the grid nodes and the thickness of the rods [8]. As mentioned above, consideration of intra-specimen variability allows the SPSD model to achieve dosimetric equivalence with real microstructures. The measure of intra-specimen variability used is a coefficient of variation $CV_{IS}$(%). This parameter cannot be inferred from the characteristics presented in Tables 5 and 6 (SD presented reflects the intra-population range of parameters) and should be determined from other data sets. Unfortunately, very few $CV_{IS}$ measurements are available. According to published data, $CV_{IS}$ measured in T-vertebra body for Tb.Th and Tb.Sp are 48% and 43% respectively [106]; and 10% for both parameters of iliac crest [126]. For skull bones (mandible), $CV_{IS}$ for Tb.Th was estimated as 9% [127], and 7% [128,129]. The same authors estimate the $CV_{IS}$ for Tb.Sp as 5%, 14% and 25%. Table 7 presents the assumed values of $CV_{IS}$% for different bones for all ages. The values for "other bones" were obtained by averaging CVs for measured bones.

### Dimensional parameters (macro-parameters) of bone segments in the hematopoietic sites of human skeleton

For most bones, there are commonly used measured dimensional parameters. In this study, for each hematopoietic bone site, we created tables that summarized published data on standard measurements, which were then averaged according to age (S1 Femur–S14 Hand Foot). For SPSD modeling, the bone was divided into segments, and the dimensions of the

**Table 7. Assumed Tb.Th and Tb.Sp values (CV$_{IS}$%) of intra-bone variability.**

| Bone | Tb.Th CV$_{IS}$% | Tb.Sp CV$_{IS}$% |
| --- | --- | --- |
| Pelvic bone | 10 | 10 |
| Vertebrae and sacrum | 48 | 43 |
| Skull bones | 8 | 15 |
| Other bones (average value) | 22 | 23 |

identified segments were described using these commonly measured values or were derived from them. Table 8 presents the total number of BSs assumed for modeling. For each unique BS, a unique BPS was generated.

The bone segmentation has already been discussed for the example of adult vertebra [5] and rib [130], as well as femur of reference ages [6]. Bones of complex shape and heterogeneous structure have been described by several bone segments. Large bones of simple shape (e.g., flat bones of the skull, flat part of the pelvic bone, long diaphysis of tubular bones) were modeled with a small fragment with maximum linear dimensions not exceeding 30 mm; then, if necessary, the total AM mass of the entire bone was estimated.

Sex differences in the size of the BS can lead to different distributions of the bone marrow within the hematopoietic site that influences the result of dosimetric modeling (average skeletal absorbed dose rate in the target organ – AM). However, the size of a human skeleton is practically independent of sex up to the age of about 10 years as indicated, in particular, by data on the growth and size changes of human bones. For example, the 50[th] percentile and 10[th]-90[th] percentile interval

**Table 8. Number of unique BSs and corresponding BPSs describing hematopoietic sites at different ages.**

| Hematopoietic site (BS) | 0-Y | 1-Y | 5-Y | 10-Y | 15-Y | Adult |
| --- | --- | --- | --- | --- | --- | --- |
| Clavicles | 3 | 3 | 3 | 3 | 6 ($3_m + 3_f$) | 6 ($3_m + 3_f$) |
| Femora | 3 | 3 | 4 | 3 | 4 ($2_m + 2_f$) | 4 ($2_m + 2_f$) |
| Fibulae | 1 | 1 | 1 | 1 | – | – |
| Humeri | 3 | 3 | 3 | 2 | 2 ($1_m + 1_f$) | 2 ($1_m + 1_f$) |
| Pelvic bones | 4 | 7 | 9 | 9 | 11 ($5_u + 3_m + 3_f$) | 11 ($5_u + 3_m + 3_f$) |
| Ribs | 1 | 1 | 1 | 1 | 8 ($4_m + 4_f$) | 8 ($4_m + 4_f$) |
| Sacrum | 5 | 9 | 5 | 5 | 15 ($5_u + 5_m + 5_f$) | 15 ($5_u + 5_m + 5_f$) |
| Scapulae | 3 | 3 | 3 | 3 | 3 | 3 |
| Skull | 1 | 1 | 1 | 1 | 1* | 1* |
| Sternum | – | – | 1 | 1 | 1 | 4 ($2_m + 2_f$) |
| Tibiae | 3 | 3 | 3 | 2 | – | – |
| Ulnae and radii | 2 | 2 | 2 | – | – | – |
| Vertebrae-C | 1 | 1 | 1 | 1 | 3 | 4 ($2_u + 1_m + 1_f$) |
| Vertebrae-T | 1 | 1 | 3 | 3 | 5 | 6 ($4_u + 1_m + 1_f$) |
| Vertebrae-L | 1 | 1 | 3 | 3 | 5 | 7 ($3_u + 2_m + 2_f$) |
| Wrist, ankle, foot and hand | 2 | – | – | – | – | – |
| Sum | 34 | 39 | 43 | 38 | 64 | 71 |
| Total | 289 | | | | | |

"-" hematopoietic site was not modeled; m – male, f – female; u – unisex;

* the data we collected do not indicate the sex differences in the skull thickness; the head circumference is larger in men than in women, but this parameter is not used in SPSD modeling.

BS = bone segment.

BPS = basic (bone) phantom segment.

of body height for 10-year-old boys and girls in Russia are 137.8 (118–156.9) cm and 138.6 (119.4–157.8) cm, respectively [131], thus, sex differences in bone size and cortical thickness were taken into account for adults and, to a certain extent, for adolescents of 15 years (Table 8). As indicated in a fundamental review [35], the average ratio of male/female bone-sizes for adults (from one population) is about 100/92, i.e., bone size for women is about 92% of the corresponding value for men. For different parts of the skeleton, this proportion may be different. Usually, for one regional population, statistical distributions of bone sizes for men and women significantly overlap including pelvic bones with obvious sex differences. As a result, a significant part of the BSs for adults and 15-Y were modeled based on combined data for male and female.

## Segmentation and parameter evaluation on the example of scapula

Fig 5 exemplifies the age dynamics of several measured parameters of the scapula, as well as the values adopted in the SPSD model (the values of all measured parameters are presented in S12 Scapula). As seen in Fig 5, the scapula characteristics have different growth dynamics. The most studied are the length and width of the scapula. These parameters are characterized by rapid growth during the first year with subsequent deceleration. A two-fold increase is observed from 0-Y to about 5-Y. In contrast, the width of the acromion does not change significantly in the first year of life; a two-fold growth is achieved by the age of 15 years. Note that data on scapula length and width (breadth) for adults were not included, because they were not used in assessing the BS parameters. The reviewed data indicate there are no sex differences in the size of various parts of the scapula.

Fig 6 illustrates the SPSD model segmentation of the scapula. The *acromion* process was modeled by a box and its width $W_a$ (shown in Fig 5C) is one of the measured parameters. The *glenoid* forms the articular surface for the shoulder joint; it was modeled by a regular cylinder with an elliptical base, and the glenoid length $D_{1g}$ (Fig 5D) is the larger diameter. The height of the glenoid cylinder for pre-adults was estimated as 16–18% of the scapula width (Fig 5B); for adults, the direct measurements of glenoid dimensions were used. The *scapular body* has been modeled for ages 0–1 years (Fig 6) by a box with an area of 30 × 30 mm and a height corresponding to the thickness of the scapula body ($S_{bh}$). The scapula length and width shown in Fig 5A, 5B were used to estimate the total area of the largest face of the scapula body (using the *IpSquare v5.0* software). Obtained values were needed to assess the weight contribution of scapula-body AM to total scapula AM. As has already been stated (Table 3), after birth, the central part of the scapula body gradually loses AM; active hematopoietic sites persist mainly in the lateral margin, which is modeled by a box (Fig 6) for ages 5-Y and older. The length of the lateral margin was taken as 85% of scapula length (Fig 5A), based on image analysis. A more detailed description of scapula data is presented in S12 Scapula. Note that the coracoid process, represented in children by a separate ossification center, as well as the spine of the scapula (*spina scapulae*) were not modeled since they contain an insignificant amount of AM.

Table 9 summarizes the many parameter values for scapula BPSs. As seen, 9–11 parameters and their SD were determined for a reference age. Most values are based on the direct scapula-measurements; others can be easily derived from them. As for Ct.Th, we found the published data for adults only; and the Ct.Th for pre-adults was estimated as a fraction of the adult-values considering that Ct.Th increases with age. To ensure that our Ct.Th estimates for pre-adults are realistic, we also analyzed scapula CT-images available on the Internet. In general, published data allow us to describe the scapula for the SPSD model by a few segments of a simple shape with parameters corresponding to average population values.

## Segmentation and parameter evaluation for other bones

The segmentation of the *long bones of the extremities, pelvic bones, sternum and clavicle* was performed in the same way as the scapulae (S1 Femur, S2 Humeri, S3 Pelvis, S5 Tibia, S8 Fibula, S10 Radius Ulna, S11 Sternum and S13 Clavicle). Bones of the *cranial vault (skull)* were modeled by one BPS (box of 30 × 30 mm area), the width (breath) and cortical

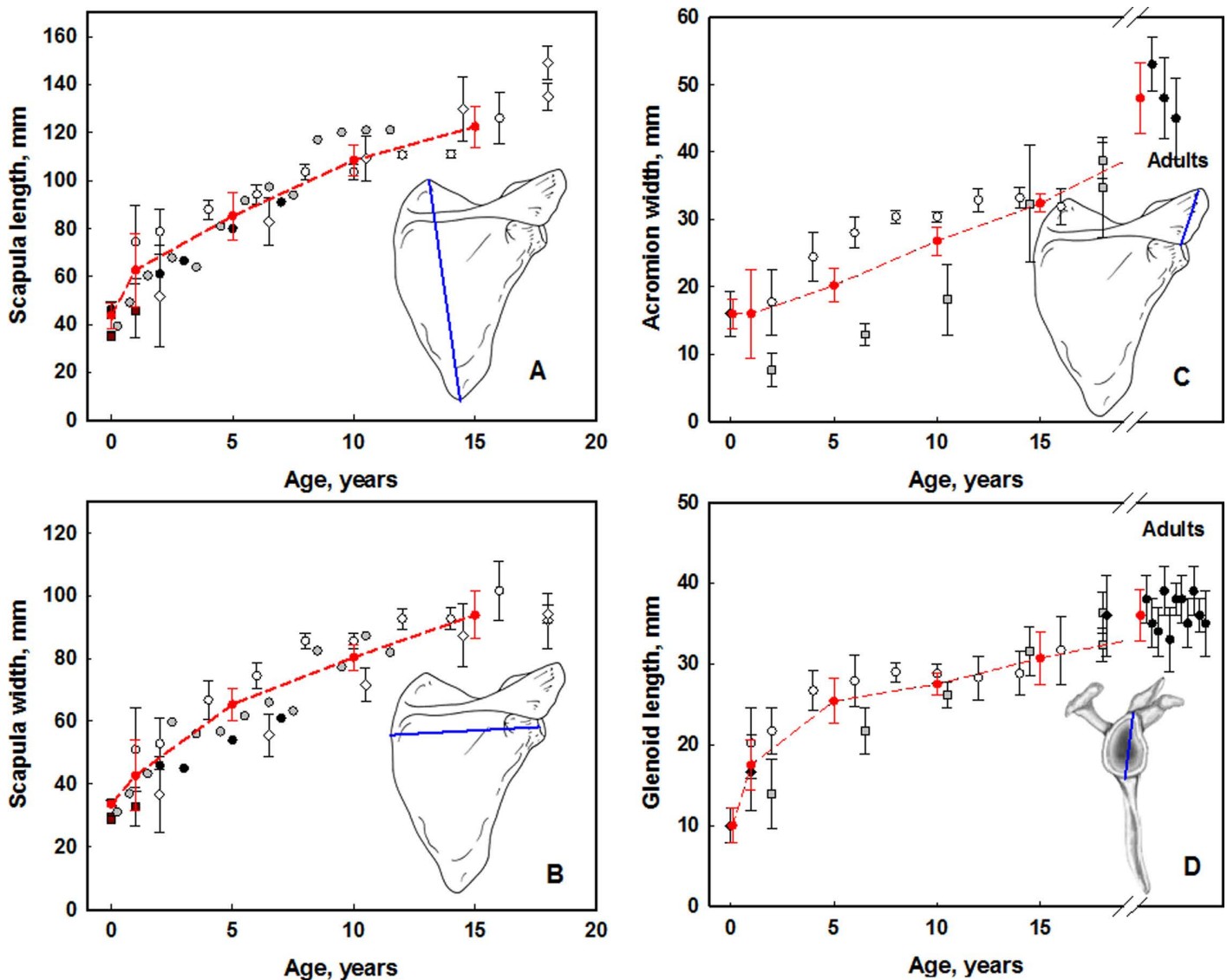

**Fig 5. Published data on scapula sizes and values assumed in SPSD model. A-** Scapula length ($S_l$): distance between the superior and inferior angles of the scapula. **B –** Scapula width (syn. Breadth; $S_w$): distance between the margin of the glenoid fossa and the medial end of the spine. **C-** Acromial width ($W_a$): maximum distance between the anterior and posterior borders of the acromion process, taken perpendicular to the axis of the spine. **D-** Length of glenoidal surface ($D_{1g}$): maximum distance between the superior and inferior borders of the glenoid articular surface. Bars represent the range of standard deviation. ○ - [132]; ■ .– [133–136]; ◇ .– [137]; ● .– [138]; ● .– [139]; ● .-SPSD reference values.

thickness of which were obtained by averaging the measured values for the frontal, temporal, occipital and parietal bones (S9 Skull).

Some hematopoietic sites are represented by a set of bones of similar type: *vertebrae, ribs, and small bones of the hands and wrist*. Approaches to the modeling of these particular hematopoietic sites are discussed in more detail.

*The vertebra column* (S6 Vertebra) consists of more than 25 vertebrae belonging to the following regions: cervical spine with C1-C7 vertebrae, thoracic spine with T1-T12 vertebrae, lumbar spine with L1-L5 vertebrae, and sacral vertebrae from 1 to 5. For dosimetric purposes, we modeled: one lumbar vertebra, representative of the entire lumbar region; one

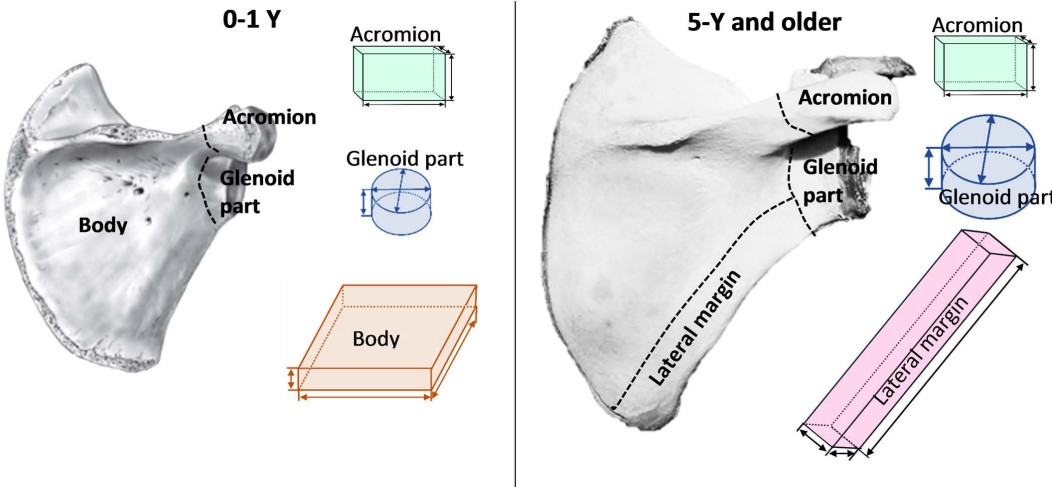

**Fig 6. Scapula segmentation illustration for ages 0-1-Y (left panel) and other ages (right panel) on the example of right scapula shown from the dorsal side.** Three segments describe the scapula at each age.

**Table 9. BS sizes and cortical thickness assumed for scapula BPSs, mm.**

| Scapula BS | Parameter | Rationale | 0-Y | | 1-Y | | 5-Y | | 10-Y | | 15-Y | | Adults | |
|---|---|---|---|---|---|---|---|---|---|---|---|---|---|---|
| | | | M | SD | M | SD | M | SD | M | SD | M | SD | M | SD |
| #1 Glenoid | $h_g$ | Pre-adults [a]:16% of $S_w$ for 0–1 Y; 18% of $S_w$ for 5–10 Y; Adults – measured values | 5.4 | 0.2 | 6.8 | 1.8 | 11.8 | 0.9 | 14.5 | 0.7 | 16.9 | 1.4 | 20 | 2 |
| #1 Glenoid | $D_{1g}$ | Measured values | 10.0 | 2.1 | 17.5 | 3.1 | 25.4 | 2.8 | 27.5 | 1.4 | 30.7 | 3.3 | 36 | 3.2 |
| #1 Glenoid | $D_{2g}$ | Measured values | 7.6 | 1.4 | 10.2 | 3.0 | 18.0 | 1.2 | 19.9 | 1.3 | 22.3 | 4.0 | 26 | 2.6 |
| #2 Acromion | $h_a$ | Based on image analysis and measured value for adults [b] | 7.0 | 1.3 | 7.0 | 1.3 | 7.6 | 1.4 | 8.2 | 1.5 | 8.8 | 1.6 | 8.8 | 1.6 |
| #2 Acromion | $W_a$ | Measured values | 16.0 | 2.2 | 16.0 | 6.6 | 20.2 | 2.5 | 26.8 | 2.1 | 32.4 | 1.3 | 48 | 5.3 |
| #2 Acromion | $L_a$ | Measured data for 0-Y and adults; for other ages [c] – 45% of measured $W_a$ | 13.0 | 3.3 | 13.0 | 5.4 | 15.7 | 1.9 | 20.9 | 1.6 | 25.2 | 1.0 | 26 | 2.9 |
| #3 Scapula body | $S_{bh}$ | 90% of measured $W_{lm}$ [d] | 2.7 | 0.35 | 2.7 | 0.35 | – | – | – | – | – | – | – | – |
| #4 Lateral margin | $W_{lm}$ | Measured values | – | – | – | – | 3.2 | 0.2 | 3.5 | 0.1 | 3.5 | 0.1 | 4.6 | 0.5 |
| #4 Lateral margin | $l_{lm}$ | Based on image analysis [e] | – | – | – | – | 10.0 | 1.2 | 10.0 | 1.2 | 10.0 | 1.2 | 10.0 | 1.2 |
| #4 Lateral margin | $h_{lm}$ | 85% [e] of measured scapular length $S_L$ | – | – | – | – | 73.5 | 8.6 | 93.5 | 5.4 | 105.6 | 7.4 | 119 | 15.5 |
| #1 | Ct.Th | Based on adults estimates: 0–1-Y- two times thinner than for adults; 5–15 years – the same as for adults | 0.45 | 0.13 | 0.45 | 0.13 | 0.9 | 0.25 | 0.9 | 0.25 | 0.9 | 0.25 | 0.9 | 0.25 |
| #2, #3, #4 | Ct.Th | | 0.4 | 0.05 | 0.4 | 0.05 | 0.8 | 0.1 | 0.8 | 0.1 | 0.8 | 0.1 | 0.8 | 0.1 |

a-assumption was taken from the images provided with a scale [20,28,140].

b-for 0-Y and 1-Y, based on images from [20]; for other ages, it is assumed that the thickness of the acromion increases uniformly and becomes equal to that for adults by the age of 15 years. Value of CV was assumed the same as in adults.

c-percentage (45% of $W_a$) was derived from measured data for 0-Y [133]. For children of 1-Y and 5-Y, CV for $L_a$ is taken equal to CV for $W_a$.

d-percentage (90% of $W_{lm}$) was derived from $S_{bh}/W_{lm}$ ratio estimated for adults (measured adult $S_{bh}$ =4.1±0.6 mm [141] (n=18); CV for children's $S_{bh}$ was taken as for $W_{lm}$ 13%.

e-from the images provided with a scale; data for pre-adults [20,28]; and adults [23,24,142].

thoracic vertebra, representative of the entire thoracic region, and one cervical vertebra, representative of C3-C7. The BSs parameters for "representative" vertebra are the values averaged over all vertebrae of the corresponding region for a specific age range. Cervical vertebrae C1-C2 for 15-Y and adults differ significantly in shape from the rest of the cervical vertebrae; therefore, they were modeled separately. The sacral vertebrae were also modeled individually, they differ significantly from each other in size, furthermore in children they are separated by cartilaginous tissue; and in adults, S-vertebrae fuse together into a single bone.

*The ribs* (S4 Rib) are also a set of bones (usually 12 pairs), they are of similar cross section but significant differences in length. In adults and 15-Y children, 4 representative rib segments were selected. Each segment was modeled by boxes of standard (maximum) length, 30 mm, but the height and width of the rib-BPSs were varied: BPS1 describes the ribs 1–2; BPS2 – ribs 3, 4, 9, 10; BPS3 – ribs 5–8; and BPS4 – ribs 11 and 12. For pre-adults of 0-, 1-, 5-, 10-Y, one rib- BPS was identified for each specific age; BPS parameters were obtained by averaging the measured data (height and width) over the whole rib-set.

The numerous bones of *wrist, ankle, hand and foot* (S14 Hand Foot) were modeled for newborns only. One cylindrical BPS was used to describe all finger phalanges of the upper and lower extremities, as well as metacarpal and metatarsal bones. For parameter estimation, the measured data of these bones were averaged. One ellipsoid BPS was used for description of ossification center of tarsus; small centers of wrist ossification were not modeled.

Note that some bone segments are paired, for example, the right and left transverse processes of vertebrae, as well as articular processes and arches. In the SPSD model, paired segments were considered equal in all parameters. Small differences in the size of the right and left limb bones were also not considered. The "pairing" of segments in hematopoietic sites as well as the multiplicity of bones within hematopoietic sites (e.g., vertebrae, rib) were considered when weighting BPS-specific DFs in accordance with the AM fraction, to obtain a skeletal average DF for $^{89,90}$Sr.

Table 8 shows sex differences are considered for 24 segments in adults and 18 segments in 15-Y. These segments describe the pelvis, proximal femur and humerus, vertebral bodies (including sacrum), clavicle, ribs, and sternum. The maximum sex differences in the BS sizes were expectedly observed for the pubic symphysis: its height and width in women was 68% and 36% less than in men respectively. Sex differences were 13–29% for the parameters of the femur-BSs, 9–12% for the humerus-BSs; 5–14% for clavicle BSs; and 15–50% for ribs-BSs.

In total, 289 unique BSs and corresponding BPSs were determined (Table 8). Note that 48.5% of BPSs are boxes, 30% - cylinders, 20% - cones of various modifications, 1.5% - prisms. The set of BPSs allows for describing the hematopoietic sites containing 90–95% of AM for all reference ages. Tables from S15 BS Sizes summarize the parameter values for each modeled BS.

## Discussion

The resolved sets of BS-parameters for six age groups (Fig 1D) are used in the "Trabecula" software [7] (Fig 1E, 1F) to generate phantoms in voxel representation, i.e., generation of basic (bone) phantom segments (BPSs). The voxel size for spongiosa depends on the thickness of the trabeculae and varies from 50 to 200 µm (average 100 µm); the same voxel sizes were used for cortical layer. Voxels describing mineralized bone (i.e., trabeculae and cortical layer) were assigned the following densities at the age of 0, 1, 5, 10, 15, and adult (>25 years): 1.65, 1.7, 1.8, 1.83, 1.85, 1.9 g/cm$^3$ respectively; for bone marrow, density is taken as 1 g/cm$^3$ for children and 0.98 g/cm$^3$ for adults. The "Trabecula" software calculates the total number of voxels of each type of tissue (cortical layer, trabeculae, bone marrow), and, therefore, evaluates the volume and mass of a BPS. If a hematopoietic site/bone is modeled by several BPSs, the total bone mass can be obtained by summing up the masses of all relevant voxels. Data on bone masses and volume were used to check the adequacy of the SPSD-model approach, since the geometric shapes used do not fully correspond to the actual shape of the bones because they are simplified/smoothed representations of the actual bone shapes. Verification of the "correctness" of bone segmentation requires sets of independent experimental data on estimates of the volumes and/or masses of wet mineralized bone.

For children, unique data on the mass of wet mineralized bones obtained by [143] in a study of 40 full-term deceased newborns were available. Before measurements, the researchers specifically removed the cartilaginous portions of the bones. The comparison of measured and SPSD-modeled bone masses is shown in Fig 7. As can be seen, the mass ranges of SPSD phantoms fall within the range of the standard deviation of the values obtained by [143], i.e., there is a good agreement of measured and modeled values.

Wang [144] studied the anatomical structure of the lumbar vertebrae in adults (n = 144). The paper does not provide the estimates of the masses, but the authors indicate the volumetric contribution of the vertebral body to the total vertebra with all processes. According to [144], the average body contribution to the total volume of male L-vertebra is 65.32 ± 4%. The segmentation of adult L-vertebra in the SPSD model was discussed in [5]; a total of 7 unique BPSs were identified for male. In such segmentation the body contribution was 67.6 ± 5%, which is not statistically different from Wang's estimates. In further work, we decided not to model vertebra pedicles due to their very small volume, we also combined two BPSs (lamina and the inferior articular process), thus 5 unique BPSs were used for L-vertebra for each sex (Table 8). The resulting volume contribution of L-vertebral body reached 70.0 ± 5%, which is also very close to Wang's estimates.

Each BPS is generated with population-averaged parameters (Fig 1E) and several (twelve) supplemental phantoms representing the limits of some parametric variabilities (Fig 1F). This approach allows modeling of the population variability of the bone size and microstructure, as well as neutralizing the model simplification (smoothing) of the bone shape. The parameters of 12 supplemental phantoms are randomly selected within the range of population variability (range of SD). As a result, the average modeled BPS occurs "inside" the set of geometric shapes of supplemental phantoms (Fig 8). When a set of DFs are calculated for BPSs and supplemental phantom segments (SPSs), their dispersion reflects the effect of variability in bone sizes and shape and microstructures.

We also compared the masses of SPSD-bone-specific phantoms with ICRP-143 [145] phantoms. The ICRP Publication provided the tissue masses for individual bones allowing calculation of total mass. Fig 9 shows SPSD-phantom masses for clavicle and pelvic bones and their variability in comparison with ICRP-143 masses. Note that the ICRP-143 approach does not imply estimates of mass variability. As can be seen in Fig 9, the masses of phantoms do not contradict each other, and the ICRP-143 values, in general, fall within the range of the standard deviations of the SPSD phantoms. In both data sources there are similar trends in sex differences, viz. in 15-year-old male phantoms, the mass of the clavicles is greater, and the mass of the pelvic bones is less than in female phantoms. As for adults, a good agreement was also observed for male and female clavicle-phantoms. However, we do not present data comparing adult pelvic bones because not all pelvic bones were described in the SPSD model (not all adult pelvic bones contain AM) (S3 Pelvis).

An important advantage of the SPSD model is the use of population-average parameters of bones and their uncertainties. Table 10 represents the number of persons whose published measurements were used; we do not give the total number of persons examined for each age because some authors measured several bones from the same person (skeleton). As expected, bone dimensions (macro-parameters) have been studied more extensively than bone microstructure, the evaluation of which requires sophisticated equipment and labor-intensive techniques.

According to Table 10, the largest number of macro-parameter measurements refers to the humerus and femur; the smallest was found for the thoracic vertebra at the age of 1-Y (1 person, several samples). We also did not find many measurements of rib height and width in children, despite the extensive data on lengths of children's ribs which were not needed in the modeling (rib-BPS represents only a rib-part of a fixed length 30 mm). Values from Table 10 reflect the availability of bone information for researchers, availability of published data for us, as well as the scientific interest in various bones often associated with the fracture frequencies. In addition, we did not use all bone macro-parameters usually measured. For example, some skull parameters important for

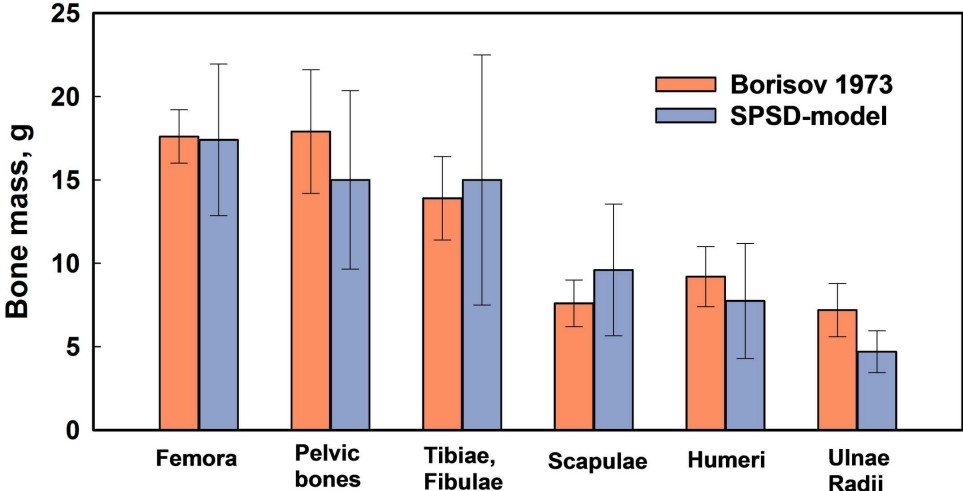

**Fig 7. Comparison of bone masses measured by Borisov [143] (n = 40) and SPSD-modeled values estimated as a sum of the masses of all BPSs describing the specific bones (left+right) of newborns considering repeated and paired segments.**

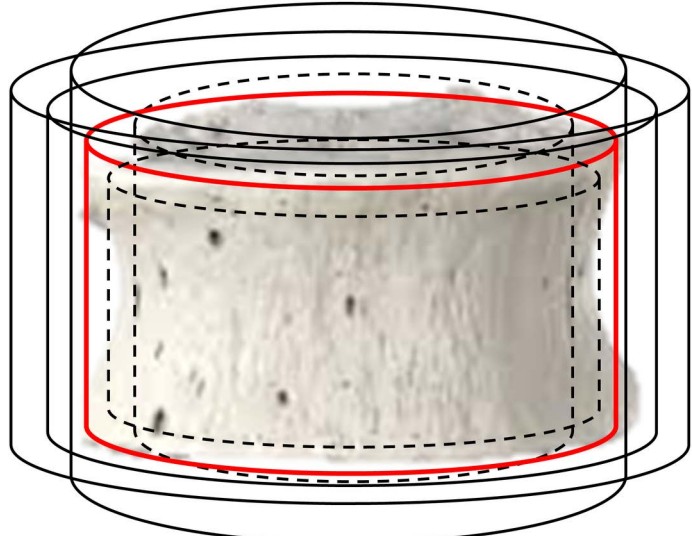

**Fig 8. Principle of BS modeling using a set of phantoms, illustration showing an example of a vertebral body.** The red line outlines the basic cylindrical phantom with population-average sizes, 5 supplemental phantoms are shown in black solid and dotted lines; their sizes were randomly selected within the limits of variability of the measured values. For the actual SPSD model, 12 phantoms are randomly generated for each bone segment; they also differ in their micro-parameters and cortical thickness.

physicians and anthropologists such as biparietal diameter and head circumference are not used for description of skull-BSs.

As for the micro-parameters (Table 10), the completeness and quality of hematopoietic data collected also vary for different ages. The most studied are the pelvic bones, mainly the ilium, because the iliac crest is the standard biopsy site to study the bone and marrow structures for medical purposes. In descending order of completeness, the list of ages

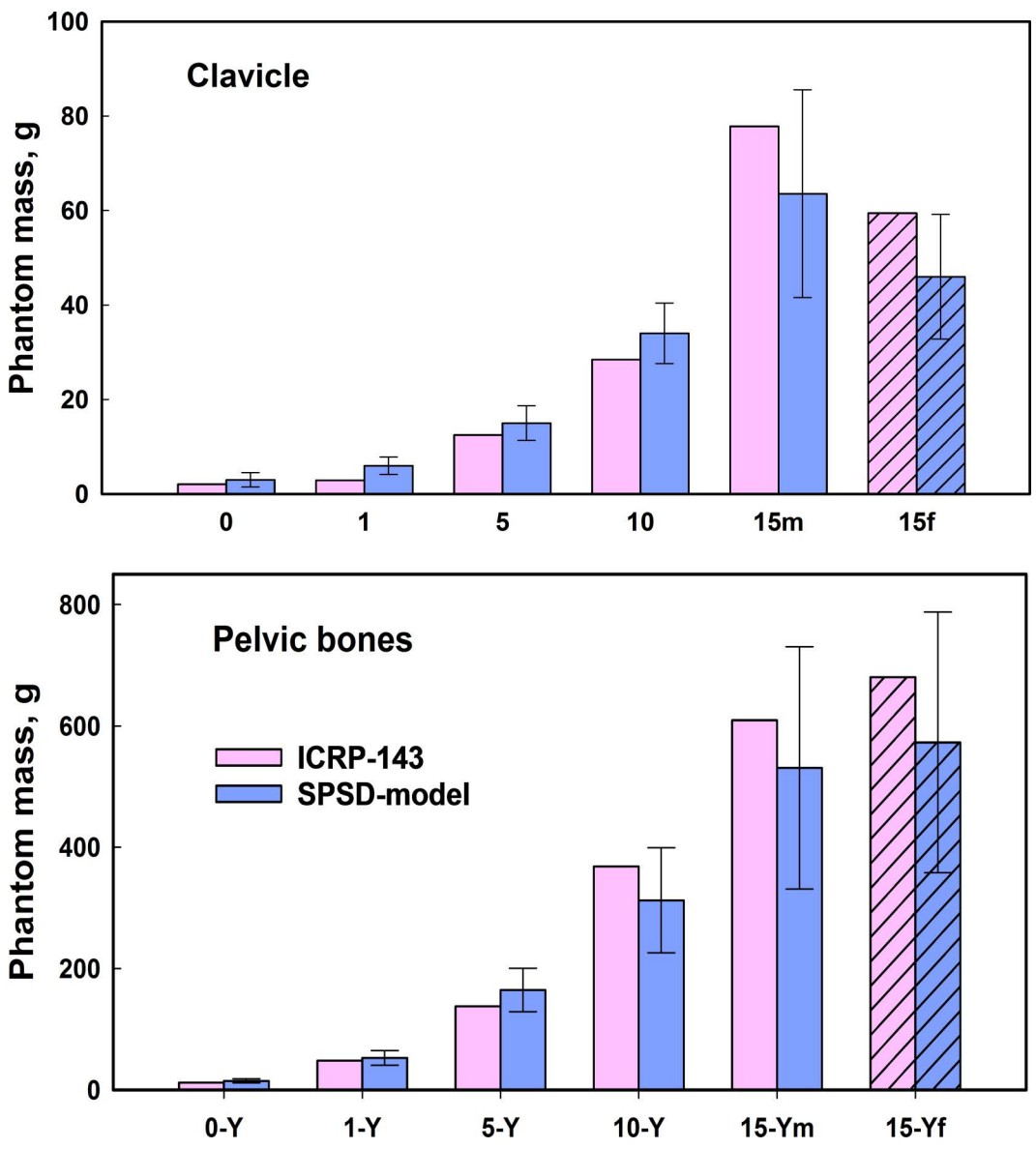

**Fig 9. Comparison of reference-age masses of ICRP-143 and SPSD phantoms simulating clavicles (left+right) – upper panel; and pelvic bones (left+right) – lower panel.** Female phantoms for 15-Y are shown with shaded bars. Vertical lines represent the range of standard deviation of SPSD-phantom masses.

would be (i) adults; (ii) infants and 1-Y-children; (iii) children and adolescents aged 5–15 years. However, the hematopoietic sites containing about 80–90% of AM are described fully for all ages (Figs 2–4). Sites for which only adult information is available (scapula, sternum and clavicle) contain about 5–8% AM [13] (10). Generally, the quality or completeness of the experimental data reflects the current level of knowledge on this matter. The data set bases will be updated as new studies are published.

As mentioned, the results of the current study (parameter values) are the input to the "Trabecula" software used to generate voxel phantoms (Fig 1E, 1F). The voxelized phantoms are used in the MCNP-6.2 code to calculate the absorbed energy in the bone marrow from the $^{89,90}$Sr source in the trabecular and cortical volumes separately, and then to calculate

**Table 10. The number of persons whose measurements were used to estimate the parameters of the BPSs in different ages.**

| Hematopoietic site | Macro-parameters | | | | | | Micro-parameters | |
|---|---|---|---|---|---|---|---|---|
| | 0-Y | 1-Y | 5-Y | 10-Y | 15-Y | Adults | 0-15-Y | Adults |
| Clavicles | 117 | 129 | 90 | 83 | 93 | 2421 | a | 45 |
| Femora | 419 | 495 | 1969 | 3532 | 2186 | 12562 | 57 | 113 |
| Fibulae+Tibiae | 1065 | 1690 | 4068 | 4208 | – | – | 42 | – |
| Humeri | 374 | 565 | 2598 | 3469 | 1749 | 1041 | 27 | 15 |
| Pelvic bones | 118 | 54 | 21 | 41 | b | 2311 | 156 | 477 |
| Radii and ulnae | 577 | 1618 | 2526 | – | – | – | 712c | 212 |
| Ribs | 10 | 8 | 7 | 16 | b | 406 | 46 | 13 |
| Sacrum | 90 | 159 | 198 | 242 | 136 | 1381 | d | d |
| Scapulae | 67 | 148 | 38 | 32 | 42 | 1889 | a | 31 |
| Skull | 55 | 87 | 87 | 82 | 64 | 405 | 18 | 174 |
| Sternum | – | – | 54 | 53 | 50 | 196 | a | 204 |
| Vertebrae-C | 47 | 16 | 54 | 113 | 177 | 1189 | 4 | 55 |
| Vertebrae-T | 70 | 1 | 33 | 16 | 19 | 786 | 68 | 126 |
| Vertebrae-L | 93 | 21 | 56 | 40 | 76 | 1940 | | |
| Wrist, ankle, foot and hand | 80 | – | – | – | – | – | 103 | – |

"-" hematopoietic site was not modeled;

a– parameters were derived from adult data and estimated age-related changes;

b– parameters were taken equal to those for adults;

c– data of HR-pQCT measurements, they were used for age-dynamics evaluation;

d– parameters were taken equal to those for combined data for T-and L-vertebrae.

DFs representing absorbed dose rate per unit concentration in mineral bone (Gy/s per Bq/kg) (Fig 1G, 1H). In turn, these DFs are averaged according to the proportion of AM in each segment and site to obtain skeletal-average DFs (Fig 1I). These separate tasks will be described in other publications.

## Conclusions

In summary, the anatomical and morphological basis for the SPSD model was created based on published data. The collection of appropriate data included the selection and analysis of original articles, atlases, manuals, monographs and the formation of primary data files. The modeling of bone segments considered the duration of active hematopoiesis and focused on large bone segments in which conversion from hematopoietic to fatty marrow occurs in pre-adulthood. Data on age-related changes in bone microstructure and on the linear sizes of human bones and their segments were analyzed.

A description of the full set of dosimetry model parameters for newborns, children 1, 5 and 10 years of age, as well as for adolescents 15 years of age and adults was presented; for the latter, sex differences in bone sizes were considered. Overall, the SPSD model includes 289 unique basic (bone) phantom segments, each of which was described by 7 or more parameters (at least three-dimensional parameters, cortical thickness, bone volume-to-total volume ratio, trabecular thickness, and trabecular space). For each parameter, population variability was evaluated. The approach to SPSD modeling, i.e., the use of simple geometric shapes, has been successfully verified using independent datasets of bone masses and volumes. The modeling results, i.e., skeletal average dosimetric factors for $^{89,90}$Sr and their uncertainties, will be the subject of a separate manuscript.

## Supporting information

**S1 Femur.** File gives a complete description of published data collected, the rationale for the choice of bone-segment parameters; and assumed SPSD-model parameters for femur.
(DOCX)

**S2 Humeri.** File gives a complete description of published data collected, the rationale for the choice of bone-segment parameters; and assumed SPSD-model parameters for humeri.
(DOCX)

**S3 Pelvis.** File gives a complete description of published data collected, the rationale for the choice of bone-segment parameters; and assumed SPSD-model parameters for pelvis (iliac, ischial, pubic bones).
(DOCX)

**S4 Rib.** File gives a complete description of published data collected, the rationale for the choice of bone-segment parameters; and assumed SPSD-model parameters for rib.
(DOCX)

**S5 Tibia.** File gives a complete description of published data collected, the rationale for the choice of bone-segment parameters; and assumed SPSD-model parameters for tibia.
(DOCX)

**S6 Vertebra.** File gives a complete description of published data collected, the rationale for the choice of bone-segment parameters; and assumed SPSD-model parameters for vertebrae (cervical, thoracic and lumbar).
(DOCX)

**S7 Sacrum.** File gives a complete description of published data collected, the rationale for the choice of bone-segment parameters; and assumed SPSD-model parameters for sacral vertebrae.
(DOCX)

**S8 Fibula.** File gives a complete description of published data collected, the rationale for the choice of bone-segment parameters; and assumed SPSD-model parameters for fibula.
(DOCX)

**S9 Skull.** File gives a complete description of published data collected, the rationale for the choice of bone-segment parameters; and assumed SPSD-model parameters for skull bones.
(DOCX)

**S10 Radius Ulna.** File gives a complete description of published data collected, the rationale for the choice of bone-segment parameters; and assumed SPSD-model parameters for radius and ulna.
(DOCX)

**S11 Sternum.** File gives a complete description of published data collected, the rationale for the choice of bone-segment parameters; and assumed SPSD-model parameters for sternum.
(DOCX)

**S12 Scapula.** File gives a complete description of published data collected, the rationale for the choice of bone-segment parameters; and assumed SPSD-model parameters for scapula.
(DOCX)

**S13 Clavicle.** File gives a complete description of published data collected, the rationale for the choice of bone-segment parameters; and assumed SPSD-model parameters for clavicle.
(DOCX)

**S14 Hand Foot.** File gives a complete description of published data collected, the rationale for the choice of bone-segment parameters; and assumed SPSD-model parameters for wrist, ankle, foot and hand bones.
(DOCX)

**S15 BS Sizes.** The excel-file contains tables formed by age with micro- and macro-parameters (sizes) and cortical thickness of each modeled bone segment (BS).
(XLSX)

## Acknowledgments

The authors acknowledge the outstanding contribution of Dr. M. Degteva (deceased 2022) to the organization and supervision of the study. The authors are sincerely grateful to Dr. Derek Jokisch (Francis Marion University) for helpful comments and discussions. The authors are grateful to D. Parshkova for her help with collecting and analyzing the published data. The authors are grateful to Dr. I. Teleshova for the opportunity to get acquainted with the anatomical collection of the South Ural State Medical University. The authors are grateful to Dr. T. Shilkova for the opportunity to study the anatomical collection of the South Ural State Humanitarian Pedagogical University (Department of General Biology and Physiology).

## Author contributions

**Conceptualization:** Evgenia I. Tolstykh, Elena A. Shishkina.

**Formal analysis:** Evgenia I. Tolstykh, Pavel A. Sharagin.

**Funding acquisition:** Michael A. Smith, Bruce A. Napier.

**Investigation:** Pavel A. Sharagin, Alexandra Yu Volchkova, Michael A. Smith.

**Methodology:** Evgenia I. Tolstykh, Elena A. Shishkina.

**Project administration:** Michael A. Smith, Bruce A. Napier.

**Software:** Pavel A. Sharagin.

**Supervision:** Michael A. Smith, Bruce A. Napier.

**Writing – original draft:** Evgenia I. Tolstykh, Bruce A. Napier.

**Writing – review & editing:** Michael A. Smith.

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
