## [Decision Letter · Decision Letter 0]

Dear Dr. Smith,

Thank you for submitting your manuscript to PLOS ONE. After careful consideration, we feel that it has merit but does not fully meet PLOS ONE’s publication criteria as it currently stands. Therefore, we invite you to submit a revised version of the manuscript that addresses the points raised during the review process.

We look forward to receiving your revised manuscript.

Kind regards,

Hesham M.H. Zakaly, Ph.D.

Academic Editor

PLOS ONE

 [This work was funded by Federal Medical-Biological Agency of Russia (MOD, EIT, EAS, PAS, VIZ) and the U.S. Department of Energy’s Office of International Health Programs (MAS, BAN) in the framework of joint US-Russia JCCRER Project 1.1 (https://www.energy.gov/ehss/russian-health-studies-program).]. 

3. Please include a caption for figure 5, 6, 7, 8.

Additional Editor Comments (if provided):

Reviewers' comments:

Reviewer's Responses to Questions

**Comments to the Author**

1. Is the manuscript technically sound, and do the data support the conclusions?

Reviewer #1: Yes

Reviewer #2: Partly

Reviewer #3: No

2. Has the statistical analysis been performed appropriately and rigorously?

Reviewer #1: Yes

Reviewer #2: No

Reviewer #3: I Don't Know

3. Have the authors made all data underlying the findings in their manuscript fully available?

Reviewer #1: Yes

Reviewer #2: Yes

Reviewer #3: Yes

4. Is the manuscript presented in an intelligible fashion and written in standard English?

Reviewer #1: Yes

Reviewer #2: Yes

Reviewer #3: Yes

Reviewer #1: the text provides a good overview of the purpose of the study, but it could benefit from explicitly stating its primary objective earlier. For example, clearly define whether the focus is methodological innovation, parameter evaluation, or both.

Reviewer #2: Dear Authors,

Thank you for your submission to PLOS ONE. Your manuscript presents a well-structured and methodologically sound study on Stochastic Parametric Skeletal Dosimetry (SPSD) for modeling radiation exposure to bone marrow. The approach is innovative and provides a robust framework for radiation risk assessment, particularly for bone-seeking radionuclides. However, I have noted some areas for improvement, particularly regarding statistical analysis, model validation, and clarity in the discussion of limitations.

Include confidence intervals (CIs) or sensitivity analysis for key parameters. The Monte Carlo approach incorporates variability, but confidence intervals for trabecular thickness (Tb.Th), bone volume fraction (BV/TV), and cortical thickness (Ct.Th) are not explicitly reported. Providing 95% confidence intervals, standard deviations, or a bootstrapping approach would improve statistical robustness.

Perform a sensitivity analysis to evaluate how variations in key parameters influence dose estimations. A one-at-a-time (OAT) sensitivity analysis or variance-based sensitivity analysis (e.g., Sobol indices) could provide valuable insights into parameter impact.

Provide empirical validation or benchmark comparisons. The study relies on literature-based data, but additional validation would strengthen credibility. If direct experimental validation is not possible, consider benchmarking results against ICRP reference phantoms or other computational dosimetry models (e.g., UF/NCI hybrid phantoms). A comparison table summarizing differences in dose estimations would be helpful.

Verify assumptions on parameter distributions. The manuscript assumes normal distributions for some trabecular bone parameters but does not test this assumption. A Kolmogorov-Smirnov (K-S) test or Shapiro-Wilk test should be performed to confirm normality. If non-normal, a log-normal or gamma distribution may be more appropriate.

Clarify segmentation methodology. The segmentation of hematopoietic sites into simple geometric shapes is described, but the rationale behind specific segmentations is not clear. A brief explanation of why certain segmentations were chosen, particularly for heterogeneous bone structures, would improve transparency.

Improve readability in the methodology section. Some descriptions, especially in parameter selection and stochastic modeling, are complex and may be difficult for non-specialists to follow. Simplifying descriptions and adding a schematic diagram or flowchart of the SPSD modeling workflow would enhance clarity.

Address minor grammatical issues. While the manuscript is generally well-written, refining sentence structures and improving conciseness would improve readability.

These revisions will improve the statistical rigor, validation strength, and clarity of the manuscript. Addressing these concerns will enhance the study’s impact and ensure its findings are transparent and reproducible. Looking forward to the revised version.

Reviewer #3: Thank you for massive data you gathered and the manuscript, but I didn't understand your introduction, your method and finally your discussion and I didn't find coordination between different parts of your manuscript. your abstract can not convey your work and is too long.

generally it is not obvious that who will use this model and how. I think it needs total and basic rewriting and revision.

**Do you want your identity to be public for this peer review?** For information about this choice, including consent withdrawal, please see our Privacy Policy

Reviewer #1: No

Reviewer #2: No

Reviewer #3: No

---

## [Author Response · Author response to Decision Letter 1]

28 Mar 2025

Detailed responses have been provided in uploaded Word file "Responses to reviewers_20250327". Thank you.

---

## [Decision Letter · Decision Letter 1]

Stochastic parametric skeletal dosimetry model for humans: Anatomical-morphological basis and parameter evaluation

PONE-D-24-52145R1

Dear Dr. Smith,

We’re pleased to inform you that your manuscript has been judged scientifically suitable for publication and will be formally accepted for publication once it meets all outstanding technical requirements.

Kind regards,

Hesham M.H. Zakaly, Ph.D.

Academic Editor

PLOS ONE

Additional Editor Comments (optional):

Reviewers' comments:

Reviewer's Responses to Questions

**Comments to the Author**

Reviewer #2: All comments have been addressed

Reviewer #3: All comments have been addressed

2. Is the manuscript technically sound, and do the data support the conclusions?

Reviewer #2: Yes

Reviewer #3: Yes

3. Has the statistical analysis been performed appropriately and rigorously?

Reviewer #2: Yes

Reviewer #3: Yes

4. Have the authors made all data underlying the findings in their manuscript fully available?

Reviewer #2: Yes

Reviewer #3: (No Response)

5. Is the manuscript presented in an intelligible fashion and written in standard English?

Reviewer #2: Yes

Reviewer #3: Yes

Reviewer #2: I have reviewed the authors’ responses and revised manuscript. The authors have addressed the key concerns appropriately, either by implementing the requested changes or by providing clear and reasonable justifications where direct implementation was not feasible within the scope of this paper. The manuscript is now well-structured, methodologically sound, and clearly defines its contribution. I recommend the manuscript for acceptance.

Reviewer #3: (No Response)

**Do you want your identity to be public for this peer review?** For information about this choice, including consent withdrawal, please see our Privacy Policy

Reviewer #2: **Yes: ** Mert Ocak

Reviewer #3: No

---

## [Editor Report · Acceptance letter]

PONE-D-24-52145R1

PLOS ONE

Dear Dr. Smith,

I'm pleased to inform you that your manuscript has been deemed suitable for publication in PLOS ONE. Congratulations! Your manuscript is now being handed over to our production team.

Kind regards,

on behalf of

Dr. Hesham M.H. Zakaly

Academic Editor

PLOS ONE